# A pathogenic human Orai1 mutation unmasks STIM1-independent rapid inactivation of Orai1 channels

**Priscilla S-W Yeung[†], Megumi Yamashita[†], Murali Prakriya\***

Department of Pharmacology, Northwestern University, Chicago, United States

**Abstract** Ca$^{2+}$ release-activated Ca$^{2+}$ (CRAC) channels are activated by direct physical interactions between Orai1, the channel protein, and STIM1, the endoplasmic reticulum Ca$^{2+}$ sensor. A hallmark of CRAC channels is fast Ca$^{2+}$-dependent inactivation (CDI) which provides negative feedback to limit Ca$^{2+}$ entry through CRAC channels. Although STIM1 is thought to be essential for CDI, its molecular mechanism remains largely unknown. Here, we examined a poorly understood gain-of-function (GOF) human Orai1 disease mutation, L138F, that causes tubular aggregate myopathy. Through pairwise mutational analysis, we determine that large amino acid substitutions at either L138 or the neighboring T92 locus located on the pore helix evoke highly Ca$^{2+}$-selective currents in the absence of STIM1. We find that the GOF phenotype of the L138 pathogenic mutation arises due to steric clash between L138 and T92. Surprisingly, strongly activating L138 and T92 mutations showed CDI in the absence of STIM1, contradicting prevailing views that STIM1 is required for CDI. CDI of constitutively open T92W and L138F mutants showed enhanced intracellular Ca$^{2+}$ sensitivity, which was normalized by re-adding STIM1 to the cells. Truncation of the Orai1 C-terminus reduced T92W CDI, indicating a key role for the Orai1 C-terminus for CDI. Overall, these results identify the molecular basis of a disease phenotype with broad implications for activation and inactivation of Orai1 channels.

**\*For correspondence:**
m-prakriya@northwestern.edu

[†]These authors contributed equally to this work

## Editor's evaluation

This manuscript provides strong evidence on the molecular basis of a mutation (L138F) in Orai1 channels that is associated with tubular aggregate myopathy. This disease-related mutation results in a gain of function of Orai1 channels due to a steric clash clash between TM1 and TM2. The study further suggests that Ca$^{2+}$-dependent Inactivation (CDI) is an intrinsic feature of Orai1 channels and that STIM1 fine-tunes CDI.

## Introduction

Store-operated Ca$^{2+}$ release-activated Ca$^{2+}$ (CRAC) channels formed by the Orai1 protein mediate a wide range of Ca$^{2+}$-dependent cellular processes including cell motility, proliferation, and differentiation (*Prakriya and Lewis, 2015*; *Emrich et al., 2021*). Orai1 channels open in response to release of endoplasmic reticulum (ER) Ca$^{2+}$ stores and function as an essential mechanism for Ca$^{2+}$ entry in most cells. The coupling of ER Ca$^{2+}$ stores to the opening of Orai1 channels is tightly controlled by the ER Ca$^{2+}$ sensor protein STIM1, which functions as the activating ligand for Orai1 channels (*Prakriya and Lewis, 2015*). Orai1 channels consist of six subunits forming three layers of transmembrane domains (TMs), with the pore-lining TM1 domains encapsulated by the TM2/3 ring which is in turn surrounded by TM4 helices (*Hou et al., 2012*). Although much of the early attention was focused on the pore-forming TM1 helix, it is increasingly clear that TMs 2–4 are not merely 'bystanders' intended for

structural support for the pore, but instead play an active and crucial role in relaying the STIM1 binding signal from the peripheral C-terminus to open the channel gate (*Zhou et al., 2016*; *Frischauf et al., 2017*; *Yeung et al., 2018*; *Yeung et al., 2020*; *Tiffner et al., 2021*).

Several types of loss-of-function (LOF) mutations within Orai1 or STIM1 can block channel gating and abrogate store-operated $Ca^{2+}$ entry. In human patients, these defective channels have deadly consequences, causing severe combined immunodeficiency, autoimmunity, and ectodermal dysplasia (*Feske, 2010*; *Lacruz and Feske, 2015*), highlighting the essential role of Orai1 channels for immunity and host defense. In addition, many pathogenic gain-of-function (GOF) Orai1 mutations have also been identified that cause Orai1 to open independently of STIM1 (*Nesin et al., 2014*; *Endo et al., 2015*; *Garibaldi et al., 2016*; *Böhm et al., 2017*). The ensuing chronically elevated intracellular $Ca^{2+}$ levels in unstimulated cells causes a Stormorken-like syndrome with tubular aggregate myopathy (TAM) and additional accompanying symptoms specific to the individual mutations (*Nesin et al., 2014*; *Endo et al., 2015*; *Garibaldi et al., 2016*; *Böhm et al., 2017*). Potential pore opening mechanisms of several GOF human mutations have been proposed (*Zhang et al., 2011*; *Palty et al., 2015*; *Yamashita et al., 2017*; *Yeung et al., 2018*; *Bulla et al., 2019*; *Tiffner et al., 2021*), but little is currently known about how the poorly characterized pathogenic human mutation, L138F, located in TM2 at the TM1-TM2/3 ring interface and linked to tubular aggregate myopathy causes constitutive channel activation (*Endo et al., 2015*).

In addition to high $Ca^{2+}$-selectivity and store-dependent activation, a key distinguishing feature of CRAC channels is fast $Ca^{2+}$-dependent inactivation (CDI). CDI was first described in native CRAC currents of T-cells and mast cells (*Hoth and Penner, 1993*; *Zweifach and Lewis, 1995*) and is now known to extend to all Orai isoforms (Orai1-3) when activated by STIM1 (*Lis et al., 2007*). Fast CDI serves as an important feedback mechanism for restraining Orai channel activity, limiting $Ca^{2+}$ entry at hyperpolarized potentials and preventing cellular $Ca^{2+}$ overload (*Zweifach and Lewis, 1995*; *Fierro and Parekh, 1999*). Consistent with this feedback role, mutations that limit CDI enhance the frequency of $Ca^{2+}$ oscillations and increase NFAT4 activation in HEK293 cells (*Zhang et al., 2019*). Yet, despite efforts over the last two decades, the molecular mechanism of CDI is not understood. Based on the differential effects of fast and slow $Ca^{2+}$ buffers in T cells and mast cells, CDI is known to arise from $Ca^{2+}$ microdomains around individual Orai1 channels (*Zweifach and Lewis, 1995*; *Fierro and Parekh, 1999*) with the $Ca^{2+}$ sensing site localized very close to the channel pore estimated to be ~3 nm or less from the pore (*Zweifach and Lewis, 1993*). More recent studies have indicated an essential requirement for STIM1 in mediating CDI, with the inhibitory domain (ID) of STIM1 playing a key role in the inactivation process (*Mullins et al., 2009*; *Mullins and Lewis, 2016a*). In particular, STIM1 mutants lacking the ID domain do not show CDI (*Mullins et al., 2009*), and constitutively active Orai1 mutants such as V102C and P245L do not show CDI in the absence of STIM1, but CDI is restored following co-expression of these mutants with STIM1 (*McNally et al., 2012*; *Derler et al., 2018*). Within Orai1 itself, structure-function studies have implicated the N-terminus (*Bergsmann et al., 2011*; *Mullins et al., 2016b*; *Zhang et al., 2019*), the C-terminus (*Lee et al., 2009*) as well as the intracellular TM2-TM3 loop (*Srikanth et al., 2010*) regions in CDI, but the identity of the $Ca^{2+}$ sensor, the location of the inactivation gate, and the precise role of STIM1 for CDI, including the possibility that it serves as the enigmatic $Ca^{2+}$ sensor, is poorly understood.

In this study, we addressed the molecular mechanism by which the pathogenic Orai1 mutation L138F causes constitutive channel activation. Our results indicate that L138F activates Orai1 due to steric clash with the TM1 residue T92 in the neighboring channel subunit. Consistent with this functional interaction, introduction of large amino acids (Leu, Phe, Trp) at T92 also produce very large, constitutively open Orai1 channels with high $Ca^{2+}$ selectivity. Uniquely, both L138F and T92W channels show CDI in the absence of STIM1, indicating that CDI is an intrinsic feature of Orai1 channels. However, CDI of constitutively active T92W channels exhibits higher intracellular $Ca^{2+}$ sensitivity resulting in high levels of constitutive inactivation. This altered $Ca^{2+}$ sensitivity of T92W Orai1 is 'normalized' to that of WT Orai1 channels by STIM1. Together, these findings identify a molecular phenotype with broad implications for activation and inactivation of Orai1 channels.

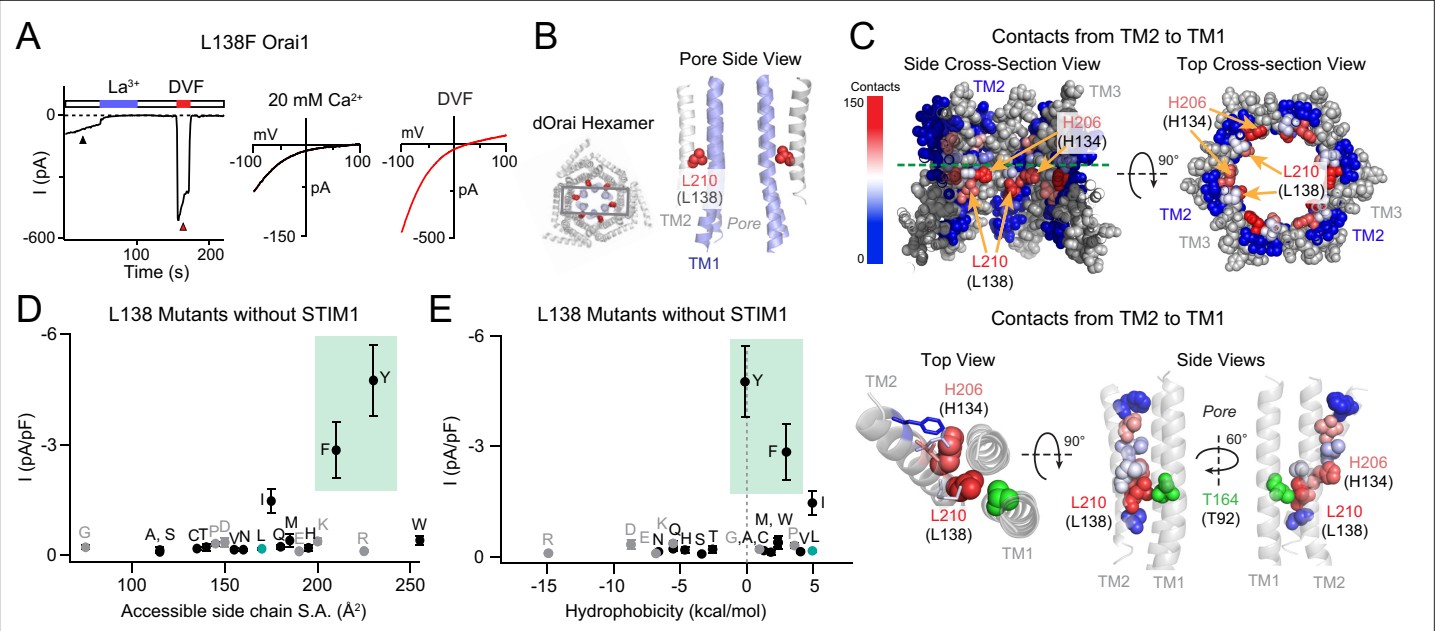

**Figure 1.** The GOF activity of L138 mutants depends on the size and shape of the introduced side chain. (**A**) Time course of the constitutively active L138F Orai1 current in the absence of STIM1. The trace shows a recording from a HEK293 cell expressing L138F Orai1. At whole-cell break-in (t=0 s), a large standing inward Ca$^{2+}$ current in 20 mM Ca$^{2+}$ external solution is observed which decreases in amplitude over the first 50 s due to current rundown. Replacing the standard external Ringer's solution with a divalent-free (DVF) solution reveals a large Na$^+$ current. The current-voltage (I-V) plots from voltage ramps (−100 mV to +100 mV) are shown on the right. (**B**) Side-view of TM2 residue L210 (hOrai1 L138, red) in the dOrai crystal structure (PDB ID: 4HKR). For simplicity, only two TM1 helices (blue) and two TM2 helices (grey) are shown. Human Orai1 numbering is shown in parentheses. (**C**) Atomic packing analysis of dOrai (PDB ID: 4HKR) showing a heat map of contacts of made by TM2 towards TM1 at the TM2/3 ring-TM1 interface. TM3 is shown in grey and TM4 is hidden for clarity. Residues are colored based on the heat map scale shown on the left that indexes the number of contacts (*Yeung et al., 2018*). In the bottom panel, TMs 1 and 2 are shown as ribbons, with TM2 residues that face TM1 shown in spheres colored according to the number of contacts made with TM1. Residue T92 is shown in green to highlight its proximity to L138. (**D–E**) Current densities of L138 mutants in the absence of STIM1 co-expression plotted against amino acid side-chain surface area and hydrophobicity. L138F/Y channels are open even without STIM1 while small and flexible substitutions produce loss-of-function channels that cannot be gated by STIM1. Green shading on these plots highlight the GOF mutants L138F and L138Y. Grey dots denote amino acids (charged or Pro/Gly) which have the potential to disrupt the helical structure of TM2. N=4–7 cells. Values are mean ± S.E.M. The intracellular solution used was the standard internal solution with 8 mM BAPTA as the Ca$^{2+}$ buffer (see Materials and methods).

The online version of this article includes the following source data and figure supplement(s) for figure 1:

**Source data 1.** Mutational analysis and contact dots analysis of the Orai1 L138 locus.

**Figure supplement 1.** Analysis of L138 Orai1 currents in the presence of STIM1.

## Results

### The TAM-linked human Orai1 L138F mutation opens the pore via steric clash with TM1

The pathogenic Orai1 L138F mutation located in TM2 was identified using whole exome sequencing in a Japanese family with tubular aggregate myopathy (TAM) and was found to mediate STIM1-independent Ca$^{2+}$ entry (*Endo et al., 2015*; *Frischauf et al., 2017*). To address the molecular mechanism of constitutive channel activation in this mutant, we overexpressed L138F Orai1 in HEK293-H cells in the absence of STIM1 and analyzed Orai1 currents by whole-cell patch-clamping. These experiments showed the presence of L138F Orai1 currents even in the absence of STIM1 co-expression. The current was consistent with Orai1 channels based on an inwardly rectifying current-voltage relationship with a reversal potential of 53.5±2.5 mV in 20 mM Ca$^{2+}$ Ringer's solution, permeation of Na$^+$ ions in divalent-free solutions, and blockade by µM concentrations of La$^{3+}$ (*Figure 1A*). Co-expression of L138F Orai1 with STIM1 resulted in store-dependent activation of Orai1 currents much larger than those evoked by L138F Orai1 alone, indicating that STIM1 can substantively enhance and further gate the activity of this mutant following store depletion (*Figure 1—figure supplement 1A and*

*B*). Interestingly, unlike many other constitutively open Orai1 mutants (*Zhou et al., 2016*; *Frischauf et al., 2017*; *Yeung et al., 2018*), Orai1 L138F showed a much larger Na$^+$ current compared to its Ca$^{2+}$ current (*Figure 1A*) (ratio of I$_{Na}$/I$_{Ca}$ was 16.1±1.9 (n=13 cells) for L138F compared to 3.5±0.6 (n=10 cells) for WT Orai1 gated by STIM1). Further, following whole-cell break-in, L138F currents exhibited rapid rundown during the first 20–30 s (*Figure 1A*). As discussed later in the paper, we believe these features are due to accumulation of calcium-dependent inactivation which limits the amplitude of the Ca$^{2+}$ current and causes Ca$^{2+}$ current rundown during hyperpolarizing pulses (see below).

How does the L138F mutation cause constitutive channel activation? The crystal structure of *Drosophila melanogaster* Orai (dOrai) in its closed state (PDB ID: 4HKR) shows that L138, which is located on TM2, protrudes towards the non-pore-lining surface of TM1 (*Hou et al., 2012*; *Figure 1B*). To understand how introduction of a bulky Phe at this position could affect channel structure, we used atomic packing analysis using a small-probe contact dots protocol to evaluate packing inter-actions of helical residues (*Word et al., 1999*; *Yeung et al., 2018*). This analysis indicated that the majority of L138 contacts are with TM1 (*Figure 1C*). In fact, of the 29 residues located in TM2, L138 displayed the highest number of contacts with TM1 (133 contacts; *Figure 1C*). H134, a residue critical to channel gating (*Frischauf et al., 2017*; *Yeung et al., 2018*) showed the second highest number of TM1 contacts (125 contacts; *Figure 1C*). Each of these residues showed substantially more contacts than other residues on TM2, which on average only had 18 contacts with TM1 (*Figure 1—source data 1*). This position of L138 and its extensive interactions with TM1 suggested that replacing the native leucine with a bulkier phenylalanine would cause steric clash with TM1 and potentially alter the conformation of TM1.

To test this possibility, we mutated L138 to several other amino acids of different sizes and shapes to assess the contribution of steric effects at this residue. Mutation of L138 to large amino acids with benzene rings such as Phe and Tyr evoked constitutively open channels that conducted ions in the absence of STIM1, whereas mutation of L138 to smaller amino acids did not affect L138 activity (*Figure 1D*, see also *Figure 1—source data 1*). Hydrophobicity did not seem to be a significant factor in the GOF phenotype (*Figure 1E*). Interestingly, mutation of the Leu at 138 to His or Trp, which are both larger than Leu, also did not significantly increase baseline Orai1 mutant channel baseline activity (*Figure 1D*), indicating that the channel is more sensitive to a benzene ring than an imidazole ring at this position. These results indicate that the introduction of a large benzene ring at L138 likely leads to steric clash of the exogenous Phe or Tyr side-chains with residues in TM1 causing channel activation. All of the constitutively active L138 mutants showed significantly larger steady-state currents when co-expressed with STIM1, indicating that the constitutively open L138 mutants can be further gated by STIM1 (*Figure 1—figure supplement 1*).

Intriguingly, analysis of mutant channel activity in the presence of STIM1 co-expression revealed that substitutions of L138 to small amino acids (G/A/S/C/T) caused loss-of-function channels that are unable to be activated by STIM1 (*Figure 1—figure supplement 1A–C* and *Figure 1—source data 1*). Mutations of L138 to polar or charged residues (N/Q/R/E/K/H) also yielded Orai1 channels with reduced currents in the presence of STIM1 (*Figure 1—figure supplement 1A, B, D*). The loss of function in these channels is not due to protein misfolding or mistargeting as measurements of E-FRET between L138A Orai1 and YFP-tagged CRAC activation domain (CAD) showed no loss of CAD binding (*Figure 1—figure supplement 1E*). Furthermore, L138E Orai1, which could not be gated by STIM1, could be strongly activated by the small molecule modulator, 2-APB, which is known to activate Orai3 and to a smaller extent Orai1 channels in the absence of STIM1 (*Yamashita et al., 2011*), indicating that the L138E mutation disrupts the STIM1-specific gating pathway without affecting channel expression or ability to conduct currents (*Figure 1—figure supplement 1D*).

Taken together, the phenotypes of the different L138 mutants show that the only substitutions retaining WT-like store-operated behavior are those that have similar hydrophobicity as the endoge-nous Leu (V/I/W/M; *Figure 1*). From the pattern of the results, we can conclude that a medium/large sized hydrophobic residue is needed at position 138 at the TM1-TM2/3 ring interface to correctly relay STIM1-dependent gating signal to the pore. The GOF phenotype of L138F human mutation appears to be driven by steric clash between the introduced benzene ring in Phe with an unknown TM1 residue to evoke channel activation.

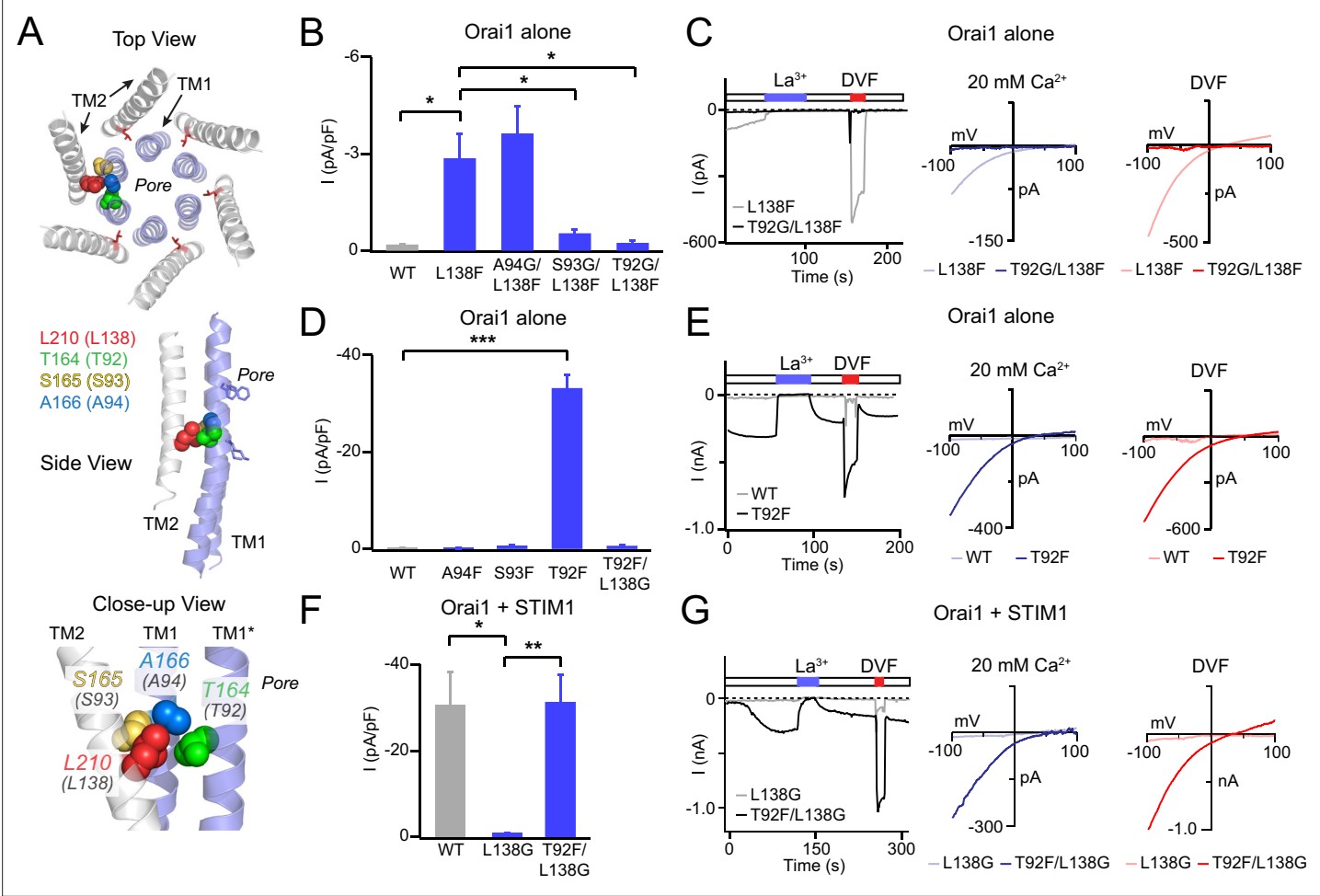

**Figure 2.** Analysis of L138-TM1 double mutants indicates that L138 interacts with T92. (**A**) Three views of L210 (hOrai1 L138, red) in the dOrai crystal structure (PDB ID: 4HKR) surrounded by the nearby TM1 residue T164 (hOrai1 T92, green) from a neighboring subunit, and S165 (hOrai1 S93, yellow) and A166 (hOrai1 A94, blue) from the same subunit. (**B**) The Orai1 L138F GOF phenotype is reversed by the glycine substitutions at T92 and S93, but not by a Gly substitution at A94. The bar graph plots the current density in the indicated mutants. (**C**) Time course and current-voltage relationships highlighting the reversal of the L138F GOF phenotype by T92G. The L138F single mutant trace and I-V plots are the same as those shown for L138F in *Figure 1A*. (**D**) Introduction of a Phe residue at T92, but not at A94 or S93 produces constitutively active GOF channels. This GOF phenotype of T92F is completely reversed by neutralizing the bulky Phe residue at 92 with a Gly substitution at L138. The bar graph shows the current densities in the indicated mutants. (**E**) Example traces of T92F or WT Orai1 currents expressed in the absence of STIM1. T92F evokes a constitutively active, $Ca^{2+}$-selective current. (**F–G**) Loss of STIM1-mediated gating of L138G is reversed by a Phe substitution at T92 (L138G/T92F mutant), suggesting that the total size chain size at this locus is crucial for channel activation. N=4–6 cells. The intracellular solution used was the standard internal solution with 8 mM BAPTA as the $Ca^{2+}$ buffer (see Materials and methods). In panels B, D, and F, values are mean ± S.E.M. *$p<0.05$ by one-way ANOVA followed by unpaired t-test between the indicated variants. **: $p<0.01$ by unpaired t-test, ***: $p<0.001$ by unpaired t-test.

The online version of this article includes the following source data and figure supplement(s) for figure 2:

**Source data 1.** Double mutants of L138 with neighboring TM1 residues reveals an interaction with T92.

**Figure supplement 1.** S93F and A94F mutations impede channel activation by STIM1.

## A screen of potential partners on TM1 reveals an L138-T92 interaction

What is the residue on TM1 that clashes with L138F to cause channel activation? In the closed state structure of *Drosophila* Orai (*Hou et al., 2012*), there are three residues on the non-pore-facing side of TM1 within 3 Å of L138 that could act as potential interaction partners for L138: S93 on the same subunit, A94 on the same subunit, and T92 from the neighboring subunit (*Figure 2A*; *Hou et al., 2012*). We hypothesized that L138F likely causes constitutive channel activation via steric hindrance with one of these three residues. To address this hypothesis, we next examined whether relieving this clash by reducing the size of the opposing residue could relieve constitutive channel activity. To test

this 'rescue' idea, we introduced a Gly at positions 92, 93, and 94 and asked if this reverses the GOF phenotype of L138F Orai1. We found that the A94G substitution did not alter the GOF phenotype of L138F (*Figure 2B*). On the other hand, introduction of Gly at the T92 position in the T92G/L138F double mutant reversed the GOF phenotype and caused the channel to be no longer constitutively active (*Figure 2B and C*). T92G/L138F Orai1 channels could still be normally activated by STIM1 with the expected properties of Orai1 channels (*Figure 2—figure supplement 1A*) indicating that the Gly substitution at T92 is unlikely to grossly disrupt Orai1 structure and function. Finally, S93G/L138F channels showed an intermediate phenotype with less activity than L138F but still higher than WT (*Figure 2B*). Together, the results from these rescue experiments suggest that the L138F Orai1 mutant is activated due to steric clash with either S93 on TM1 of the same subunit or T92 of the neighboring subunit.

If steric clash between L138F and T92 or S93 explains the GOF phenotype of L138F, we reasoned that introduction of large amino acids at these loci in Orai1 should also evoke GOF channels through steric clash with the endogenous L138 residue. We tested this hypothesis by mutating each of these three potential interaction sites on TM1 (A94, S93, and T92) sequentially to Phe, the same amino acid that evokes constitutive activation when introduced at L138 (*Figure 2D*). This analysis indicated that only the mutation of T92 to Phe produced constitutively open channels (*Figure 2D and E*). By contrast, S93F and A94F mutant channels were not constitutively active, and in fact these mutations also impeded STIM1-mediated channel function (*Figure 2—figure supplement 1B*). Importantly, as seen with the T92G/L138F rescue experiment, introduction of L138G in the strongly active T92F channel completely reversed its GOF phenotype (*Figure 2D*). Thus, the most straightforward interpretation of these results is that the L138F mutation, and correspondingly, the T92F mutation evoke tonic channel activation due to steric clash between adjacent pairs of TM1 and TM2 helices.

As noted above, small amino acids at L138 lead to LOF channels that cannot be gated by STIM1. We therefore hypothesized that this loss of gating may be rescued by restoring contacts in this region by introducing a larger side-chain at T92. Consistent with this hypothesis, the introduction of a large aromatic amino acid, Phe, and T92 in a T92F/L138G double mutant fully restored STIM1-mediated gating (*Figure 2F and G*). Together, these results indicate that steric contacts between the pore helix and TM2 in this region of the channel are essential for channel gating and finely tuned to relay the STIM1-mediated conformational change to the pore formed by TM1.

## Large amino acids mutations at T92 cause strong activation of Orai1 channels

The results presented above indicate that T92F channels are tonically active due to steric clash between T92F and L138. To address whether the constitutive channel activity of T92F shares side-chain size dependence analogous to that seen at position L138 (*Figure 1C*), we mutated T92 to other large and small amino acids and analyzed the mutant Orai1 channel currents in the absence or presence of STIM1. We found that substituting larger amino acids at this position produced GOF channels with large Orai1 currents even in the absence of STIM1 co-expression (*Figure 3A* and *Figure 3—source data 1*), whereas mutants with small, polar substitutions (G/A/S/C) did not cause constitutive activity but instead retained store-operated behavior (*Figure 3*, *Figure 3—figure supplement 1*). In general, amino acids with larger surface area (L/M/F/Y/W) produced much larger Orai1 currents (25–35 pA/pF) compared to amino acids with intermediate size (V/I/H) (5–10 pA/pF) (*Figure 3A*). The largest constitutively active currents were seen with the Orai1 T92L/M/F/Y/W mutations, which produced $Ca^{2+}$-selective currents with reversal potentials >50 mV, similar to the highly active Orai1 H134A/C/S/T and ANSGA channels described previously (*Zhou et al., 2016*; *Frischauf et al., 2017*; *Yeung et al., 2018*). Consistent with the high baseline activity of these mutants, T92L/M/F/Y/W channels were not noticeably further activated following whole-cell break-in when co-expressed with STIM1 (*Figure 3—figure supplement 1*). Moreover, these T92 mutants displayed much larger $Ca^{2+}$ currents compared to L138F/Y channels, with current densities of 30–35 pA/pF vs 3–5 pA/pF, respectively (*Figure 2* and *Figure 3*). We hypothesize that the substantially larger tonic currents elicited by T92 mutations compared to L138 mutations may be related to the size of the endogenous residue at position 92. Because the side-chain surface area of Thr (140 Å$^2$) is smaller than that of Leu (170 Å$^2$), replacing T92 with bulky residues likely produces much greater degree of steric clash at the TM1-TM2 interface and therefore stronger channel activation than the complementary L138F substitution, where the native

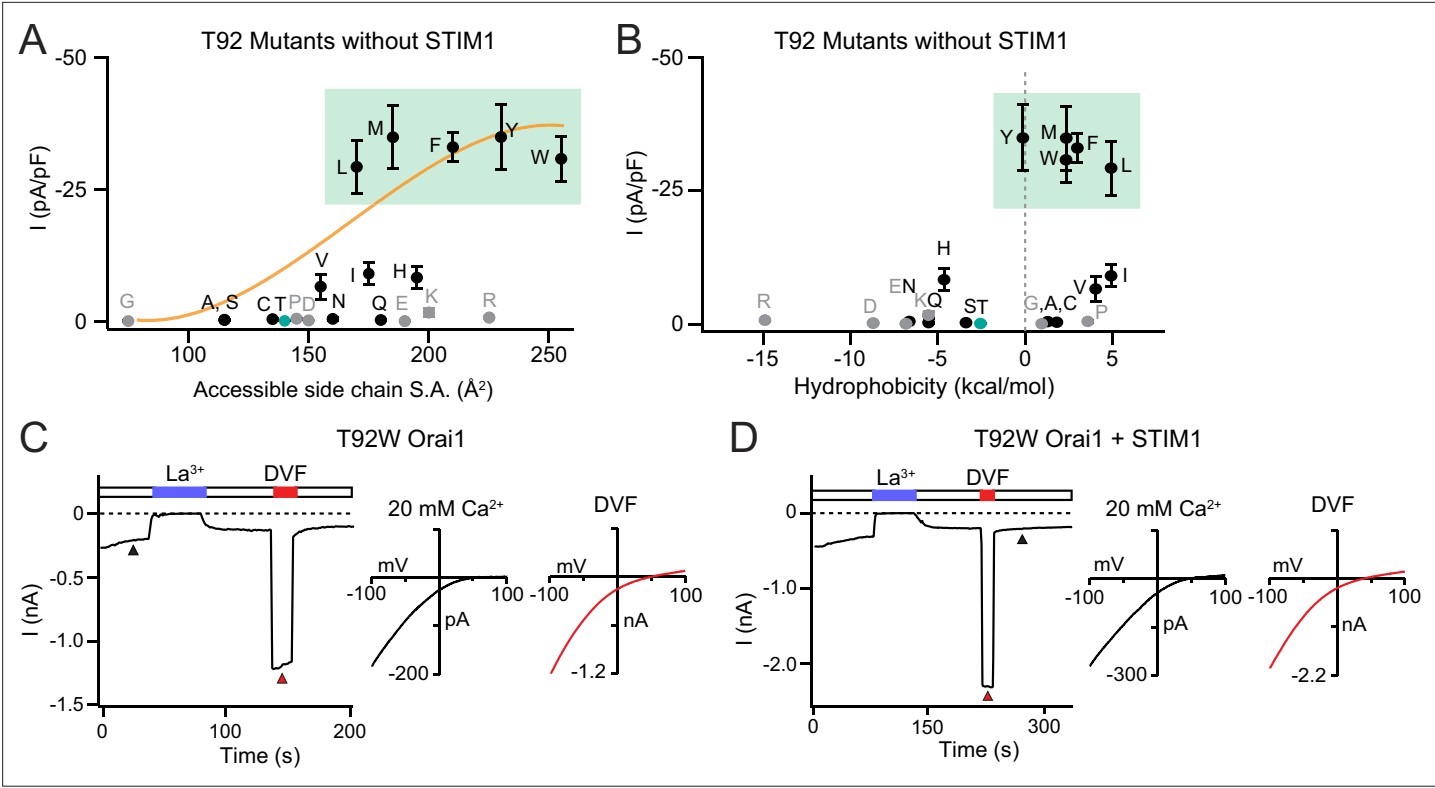

**Figure 3.** Large amino acid substitutions at T92 cause constitutive Orai1 activation. (**A–B**) The current densities of T92 mutants in the absence of STIM1 plotted against side chain size and hydrophobicity, showing that bulky substitutions at T92 cause GOF channels. Plots show peak Orai1 currents recorded at –100 mV in HEK293 cells expressing the indicated Orai1 mutants. N=4–8 cells. Values are mean ± S.E.M. The green shaded areas denote residues that evoke GOF phenotypes. Grey dots denote amino acids (charged or Pro/Gly) which have the potential to disrupt the helical structure of TM2. (**C**) Example time course of T92W Orai1 current following whole-cell break-in at t=0 s in a cell expressing T92W Orai1 alone. The I-V graphs in 20 mM $Ca^{2+}$ and DVF solutions are shown the right. T92W Orai1 alone shows inward rectifying I-V curves with positive reversal potentials, similar to STIM1-activated WT Orai1 channels. (**D**) Example time course and I-V plots of T92W Orai1 with STIM1 co-expression (Orai1:STIM1=1:5 cDNA transfection ratio). STIM1 does not significantly boost the current amplitude of GOF T92W Orai1 channels, suggesting that this mutant is nearly fully active at baseline. The intracellular solution used was the standard internal solution with 8 mM BAPTA as the $Ca^{2+}$ buffer (see Materials and methods).

The online version of this article includes the following source data and figure supplement(s) for figure 3:

**Source data 1.** Orai1 channel activity is correlated with side chain size at T92.

**Figure supplement 1.** Analysis of T92 Orai1 mutant currents in the presence of STIM1.

---

Leu is already quite large. Finally, like the charged substitutions at L138, Asp or Arg substitutions at T92 resulted in loss of function in the presence of STIM1 (*Figure 3—figure supplement 1B*), indicating that introduction of a charge at this locus inhibits channel gating.

### L138 and T92 GOF mutations exhibit calcium-dependent fast inactivation in the absence of STIM1

Fast $Ca^{2+}$-dependent inactivation (CDI) is a distinguishing feature of CRAC channels in which $Ca^{2+}$ flux through Orai1 at hyperpolarizing potentials (e.g. steps to –100 mV) causes negative feedback inhibition to inactivate channels on a timescale of tens to hundreds of milliseconds (*Hoth and Penner, 1993*; *Zweifach and Lewis, 1995*; *Fierro and Parekh, 1999*; *Prakriya and Lewis, 2015*). In the presence of a weak chelator such as EGTA, WT Orai1 channels activated by STIM1 inactivate by ~50% during 300 ms steps to –100 mV (*Figure 4A*). Fast CDI is regulated by a $Ca^{2+}$-sensing site located very close to the channel pore, estimated to be only 3–4 nm away from the pore (*Zweifach and Lewis, 1995*). Although the molecular basis of CDI remains unclear, recent studies have suggested that CDI involves functional coupling of the inactivation domain (ID) of STIM1 (*Derler et al., 2009*; *Scrimgeour et al., 2009*; *Mullins and Lewis, 2016a*) with the inner pore residues 76–91 of Orai1 (*Mullins et al., 2016b*). In line with the expected requirement for STIM1 for CDI, many GOF Orai1 mutants including

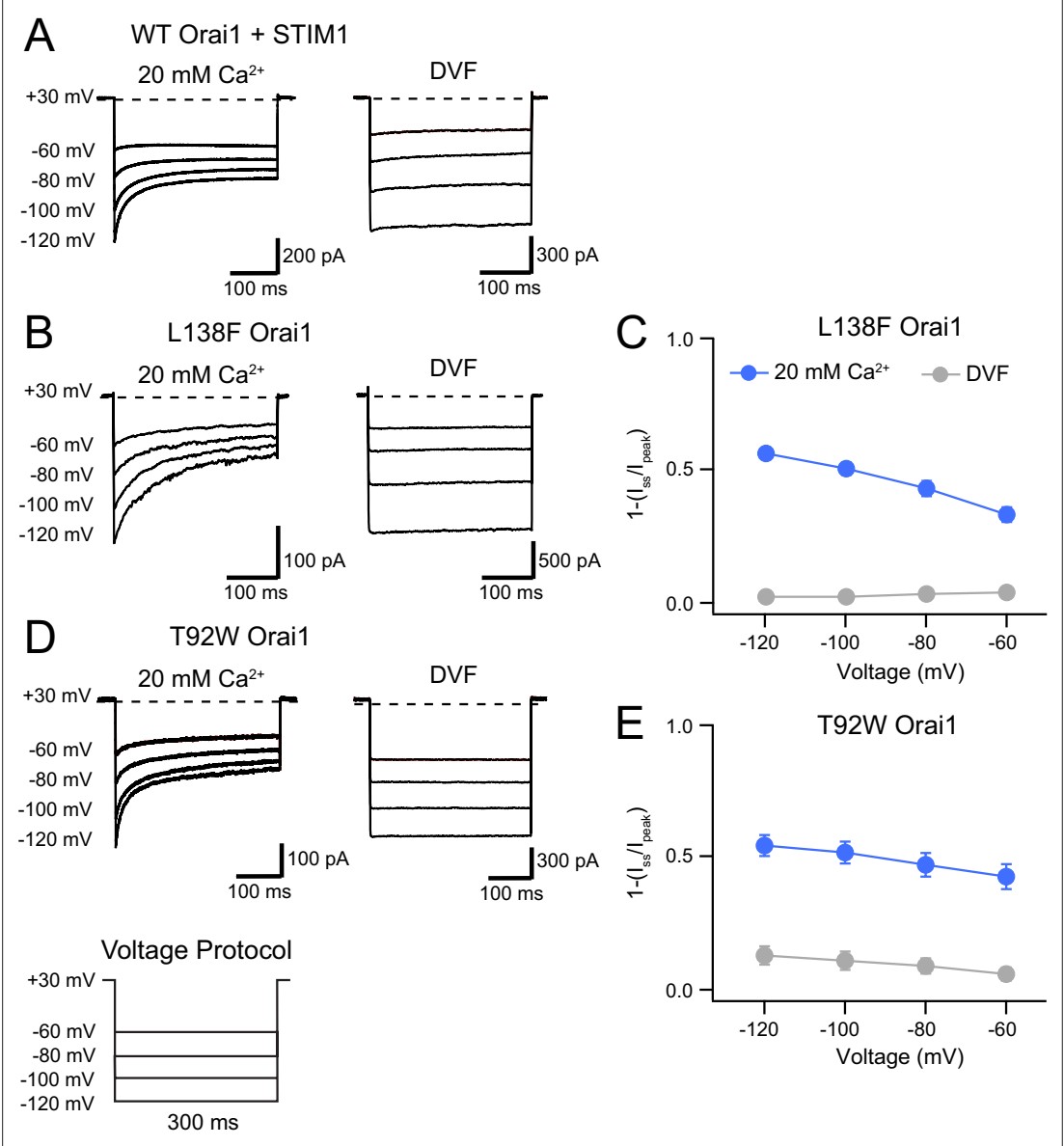

**Figure 4.** L138F and T92W Orai1 mutants show CDI. (**A**) Currents in response to hyperpolarizing voltage steps in STIM1-gated WT Orai1 channels in the presence of 10 mM EGTA in the internal solution. Hyperpolarizing voltage steps from –120 to –60 mV lasting 300ms were applied from a holding potential of + 30 mV. Replacing the extracellular 20 mM $Ca^{2+}$ Ringer's solution with a DVF solution eliminates CDI, consistent with the known calcium dependence of CDI (**Zweifach and Lewis, 1995**). (**B**) Representative traces showing CDI of L138F Orai1 currents in a cell expressing L138F Orai1 alone. The intracellular solution contained 8 mM BAPTA. (**C**) Quantification of extent of inactivation ($1-I_{ss}/I_{peak}$) of L138F GOF mutant channel without STIM1 with 8 mM BAPTA internal solution. (**D**) Representative traces of CDI of T92W Orai1 without STIM1 with an 8 mM BAPTA internal solution. (**E**) Quantification of extent of inactivation ($1-I_{ss}/I_{peak}$) of T92W Orai1 in the absence of STIM1. In both mutants, inactivation seen in 20 mM $Ca^{2+}$ is abolished in the presence of divalent-free (DVF) solution indicating that the inactivation requires conducting $Ca^{2+}$ ions. N=5–17 cells. Values are mean ± S.E.M.

The online version of this article includes the following source data and figure supplement(s) for figure 4:

**Source data 1.** Analysis of STIM1-independent inactivation in L138 and T92 mutants.

**Figure supplement 1.** Analysis of T92 mutant CDI.

**Figure supplement 2.** Kinetics of CDI in WT and T92W Orai1 channels.

H134A/S, V102C/A, and P245L do not inactivate in the absence of STIM1, but do so in the presence of STIM1 (**McNally et al., 2012**; **Zhou et al., 2016**; **Derler et al., 2018**; **Yeung et al., 2018**).

Unexpectedly and in contrast to the behavior of those GOF mutants, we observed that L138F and T92W mutants exhibited time-dependent inactivation during hyperpolarizing steps in the absence of

STIM1 (*Figure 4B–E*). In the presence of 8 mM intracellular BAPTA, hyperpolarizing steps between –60 and –120 mV caused rapid current decline in the $Ca^{2+}$ currents even in the absence of STIM1 (*Figure 4B and D*), with the current at end of a 300ms –120 mV voltage pulse declining by ~60% relative to the peak current at the beginning of the voltage step. In both mutants, inactivation was strongly reduced or absent in DVF solutions indicating that the inactivation is not mediated by $Na^+$ ions and suggesting strong divalent ion-dependence. Consistent with this interpretation, raising extracellular $Ca^{2+}$ concentration from 20 to 110 mM (isotonic $Ca^{2+}$ solution) significantly accelerated and increased the extent of T92W inactivation (*Figure 4—figure supplement 1A*). Hyperpolarizing the membrane potential to more negative potentials (e.g. –120 mV) accelerated inactivation in both mutants and increased its extent, mimicking the effects of raising extracellular $Ca^{2+}$, suggesting that inactivation of these mutants strongly depends on the single-channel current amplitude. Paradoxically, CDI in the T92W mutant is greater in BAPTA-containing internal solutions compared to EGTA-containing solutions. This unexpected behavior is described and analyzed further below.

Two-component fits of the current decay of T92W Orai1 at –100 mV in the 20 mM $Ca^{2+}$ extracellular Ringer's solution showed fast and slow $\tau$ values of 11±1ms and 134±16ms respectively, comparable to the kinetics of WT Orai1 CDI in the presence of STIM1 and with 10 mM EGTA as the internal buffer ($\tau_{fast}$ = 10 ± 1 ms and $\tau_{slow}$ = 60 ± 8 ms; *Figure 4—figure supplement 2*). Thus, the kinetics, divalent ion requirement, and dependence on driving force for $Ca^{2+}$ entry are all features qualitatively similar to CDI of CRAC channels gated by STIM1.

Interestingly, comparison of the extent of inactivation in the different constitutively active L138 and T92 mutants showed that CDI during the hyperpolarizing step is correlated with the size of the introduced side chain (*Figure 4—figure supplement 1B and C*). Specifically, although T92V was constitutively active, over the course of 100ms hyperpolarizing steps, this mutant showed no detectable inactivation but instead displayed potentiation similar to other previously described GOF mutations including V102C and H134 (*Zhou et al., 2016*; *Derler et al., 2018*; *Yeung et al., 2018*; *Figure 4—figure supplement 1B*). By contrast, larger amino acid substitutions at T92 increased inactivation with T92W exhibiting the largest amount of inactivation over the –100 mV hyperpolarizing step (*Figure 4—figure supplement 1C*). This interesting finding implies that like activation of the GOF L138F and T92W channels, inactivation may also be induced through steric clash between L138 and T92. To our knowledge, these are the first reported GOF mutants that inactivate in the absence of STIM1.

## Inactivation of T92W Orai1 exhibits aberrant dependence on intracellular $Ca^{2+}$ buffering

Previous work has shown that fast CDI is exquisitely sensitive to the species of the intracellular $Ca^{2+}$ buffer (*Zweifach and Lewis, 1995*; *Fierro and Parekh, 1999*; *Yamashita et al., 2007*). Specifically, fast CDI is not affected by the slow $Ca^{2+}$ chelator, EGTA but is strongly reduced by the fast chelator, BAPTA (*Zweifach and Lewis, 1995*; *Fierro and Parekh, 1999*), whose $k_{on}$ of $Ca^{2+}$ binding is about 400 times faster than that of EGTA. This is because although both buffers have comparable affinities for $Ca^{2+}$, the slower kinetics of $Ca^{2+}$ binding to EGTA produces a region of relatively unbuffered $[Ca^{2+}]_i$ around individual CRAC channels. By contrast, the faster $k_{on}$ of BAPTA ensures that $[Ca^{2+}]$ is strongly attenuated in the vicinity (<20 nm) of individual open CRAC channels in this chelator. Therefore, whereas BAPTA is very effective in suppressing both local and global $Ca^{2+}$ signals, EGTA mainly buffers the global $Ca^{2+}$ signal (*Neher, 1986*). We therefore used the differential effects of EGTA and BAPTA to study the local $[Ca^{2+}]_i$ dependence of T92W Orai1 inactivation.

In WT Orai1 channels activated by STIM1, we found that increasing intracellular BAPTA from 0.8 to 8 mM reduced the extent of CDI (*Figure 5A*). Because increasing the buffer concentration is predicted to markedly reduce the size of the $Ca^{2+}$ microdomain around individual CRAC channels (*Figure 5E*), this result is in line with the idea that CDI is driven by local $Ca^{2+}$ microdomains and diminishes as the local $[Ca^{2+}]$ is reduced (*Zweifach and Lewis, 1995*; *Mullins and Lewis, 2016a*). Likewise, substituting BAPTA with the slower chelator EGTA (which cannot effectively buffer local $[Ca^{2+}]$ due to its slower on-rate), increases $[Ca^{2+}]_i$ around CRAC channels (*Figure 5E*) and markedly increased the extent of CDI (*Figure 5A*). No significant change in CDI occurred by increasing intracellular EGTA from 10 to 20 mM, consistent with the known insensitivity of CDI to changes in the concentration of intracellular EGTA (*Zweifach and Lewis, 1995*) and in line with the minimal effects on local $[Ca^{2+}]_i$ by this buffer (*Figure 5E*).

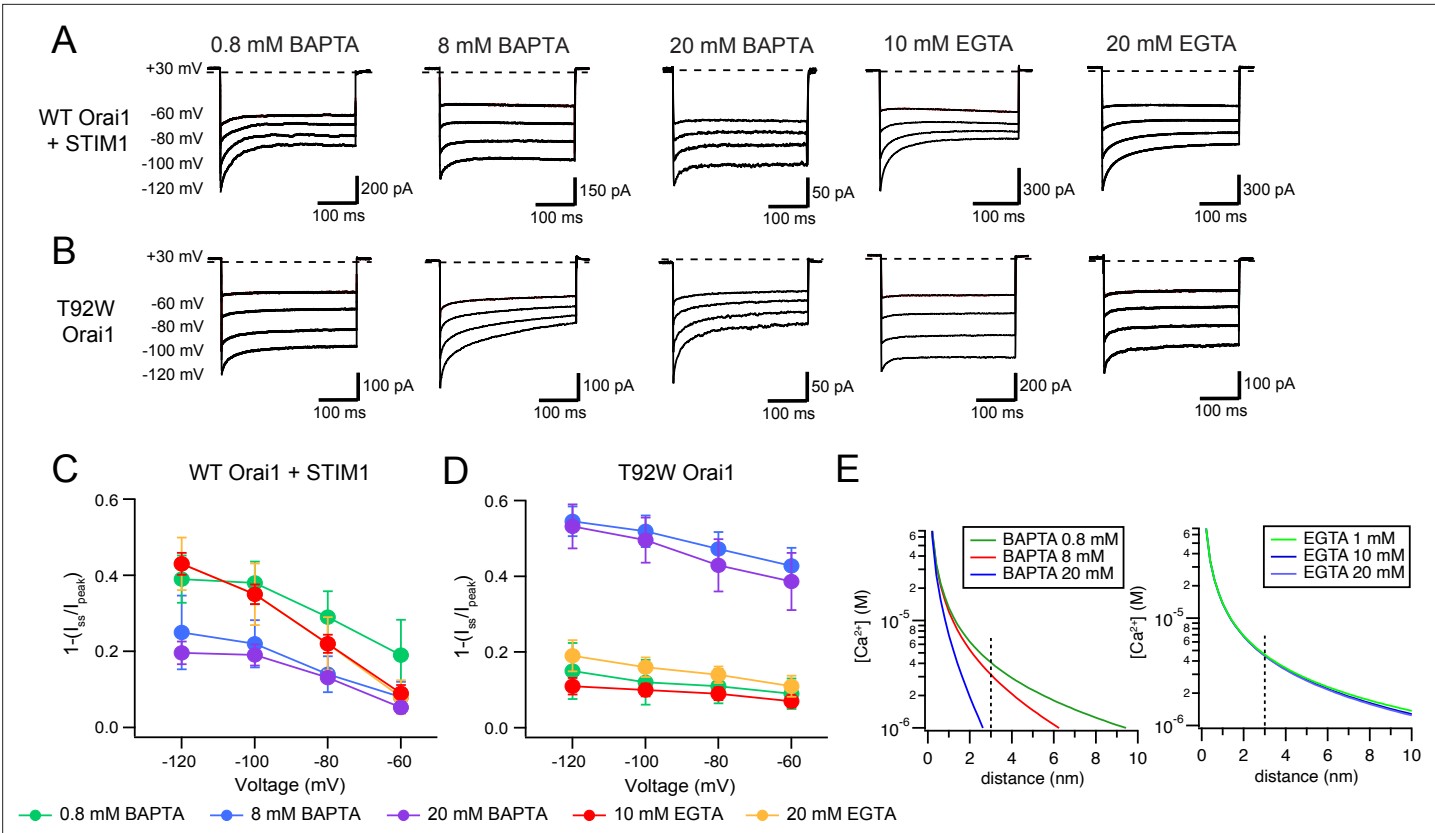

**Figure 5.** CDI of T92W Orai1 shows enhanced Ca²⁺ sensitivity compared to WT Orai1. (**A–B**) Representative inactivation traces of WT Orai1 +STIM1 and constitutively active T92W Orai1 mutant channels during voltage steps with either a slow (EGTA) or fast (BAPTA) Ca²⁺ chelator in the internal solution. (**A**) In cells expressing WT Orai1, raising the intracellular BAPTA concentration from 0.8 to 8 or 20 mM diminishes CDI. Further, replacing intracellular BAPTA with EGTA at any concentration strongly enhances CDI. (**B**) However, in T92W Orai1, increasing the BAPTA concentration from 0.8 to 8 or 20 mM enhances CDI, and replacing BAPTA with EGTA strongly diminishes CDI. (**C–D**) The extent of inactivation ($1-I_{ss}/I_{peak}$) plotted against membrane voltage in the indicated buffer solutions. The extracellular Ringer's solution contained 20 mM Ca²⁺ in all cases. N=4–17 cells. Values are mean ± S.E.M. Note that the inactivation quantification plot for T92W in 8 mM BAPTA is also shown in *Figure 4E*. (**E**) [Ca²⁺]ᵢ profiles in the presence of different concentrations of EGTA or BAPTA. [Ca²⁺] was estimated from *equation 1* (see Results). The unitary Ca²⁺ current amplitude was assumed to be 5 fA. At distances beyond 2 nm, [Ca²⁺]ᵢ is significantly buffered by BAPTA but not EGTA.

The online version of this article includes the following source data for figure 5:

**Source data 1.** Dependence of T92W inactivation on internal local Ca²⁺ concentration compared to WT Orai1 channels.

Unexpectedly, fast CDI of T92W showed very different phenotypes in response to changes in the species and concentrations of intracellular buffers. In T92W Orai1, the highest extent of CDI occurred at 20 mM BAPTA and the extent and rate of inactivation at this concentration was broadly comparable to CDI seen in WT Orai1 channels at −120 mV with 10 mM EGTA (*Figure 5B*). Reducing the concentration of intracellular BAPTA to 0.8 mM paradoxically *decreased* CDI in the constitutively active T92W mutant (*Figure 5B*), in contrast to the *increase* seen in WT Orai1 channels. Most strikingly, replacing BAPTA with 10 mM EGTA caused marked loss of CDI with only a small hint of current decay apparent during the hyperpolarizing steps (*Figure 5B*), in contrast to the behavior of WT Orai1 channels which displayed significant *increases* in CDI in EGTA (*Figure 5A*). Overall, these trends indicated that the pattern of CDI seen in response to varying the intracellular buffer (EGTA and BAPTA) is essentially reversed in the T92W mutant compared to WT Orai1 channels.

What is the explanation for this aberrant dependence of T92W Orai1 CDI on intracellular Ca²⁺ buffering? We considered and excluded several possibilities. First, we weighed whether the observed inactivation is not Ca²⁺-mediated but instead occurs due to changes in membrane voltage. However, this notion is not consistent with observations indicating that CDI is lost in the presence of Na⁺ as the current carrier (*Figure 4B and D* right traces), and that raising extracellular Ca²⁺ from 20 to 110 mM

significantly accelerates and increases the extent of CDI (*Figure 4—figure supplement 1A*). A second possibility is that inactivation of T92W Orai1 channels in EGTA containing internal solutions may occur too fast to be detected over the 300ms hyperpolarizing steps, representing some unknown type of ultra-fast activation. However, careful examination of the initial current immediately following membrane hyperpolarization revealed no such component, and increasing the current sampling rate from 5 kHz to 20 kHz failed to reveal the presence of an ultra-fast inactivating current that was missed in our recording conditions. A third possibility, that T92W disrupts (or impairs) inactivation gating is ruled out by the fact that CDI is very clearly seen in the presence of 8 and 20 mM BAPTA (*Figure 5B*) and occurs with kinetics similar to that observed for WT Orai1 channels in EGTA containing internal solutions. A fourth possibility is that CDI of the T92W still requires STIM1, with the endogenous pool of STIM1 in HEK293 cells interacting with the T92W mutant Orai1 channel with increased affinity to drive CDI. However, T92W current recordings in STIM1/STIM2 double knock-out HEK293 cells (*Emrich et al., 2019*) also showed continued persistence of CDI, indicating that the CDI of T92W occurs independently of STIM1 and STIM2 (*Figure 4—figure supplement 1D*). Finally, introducing the E106D Orai1 mutation, which abrogates CDI of heterologously-expressed Orai1 channels (*Yamashita et al., 2007*), also abrogated inactivation of T92W Orai1 currents (*Figure 4—figure supplement 1E*), confirming that the basic mechanism of inactivation gating are shared between T92W and WT Orai1 channels. Together, these results indicate that although the gating mechanism that causes inactivation is likely similar, the $Ca^{2+}$ sensing process upstream of the conformation changes that drives inactivation gating is altered in T92W mutant channels.

## CDI of constitutively active T92W Orai1 exhibits increased $Ca^{2+}$ sensitivity

One clue for why T92W Orai1 shows greater CDI in the presence of BAPTA comes from comparison of the $[Ca^{2+}]_i$ profiles that give similar amounts of CDI in different buffering conditions (*Figure 6A*). The steady-state inactivation plots show that the extent of CDI of WT Orai1 channels at −120 mV in 10 mM EGTA is quantitatively similar to CDI of T92W Orai1 at −60 mV in 8 mM BAPTA (*Figure 6A*). Diffusion models predict that the magnitude of the $Ca^{2+}$ concentration in $Ca^{2+}$ microdomains around individual Orai1 channels should be strongly influenced by the properties of the buffer and the single channel $Ca^{2+}$ current amplitude ($i_{Ca}$) (*Neher, 1986*; *Stern, 1992*; *Prakriya and Lingle, 2000*). Specifically, the steady-state $[Ca^{2+}]$ as a function of distance $d$ from a point source of $Ca^{2+}$ influx can be given by the relation:

$$[Ca^{2+}] = [Ca^{2+}]_{ss} + \frac{i_{ca}}{4\pi F d D_{Ca}} e^{\left(\frac{-d}{\lambda}\right)} \tag{1}$$

where $[Ca^{2+}]_{ss}$ is the bulk $[Ca^{2+}]_i$ (estimated to be negligible in the presence of exogenous buffers), $F$ is the Faraday's constant, $D_{Ca}$ is the diffusion constant for $Ca^{2+}$ ($3 \times 10^{-10}$ $m^2 s^{-1}$). $\lambda$, the space constant for $Ca^{2+}$ diffusion in the presence of a buffer (EGTA or BAPTA) is given by the relation:

$$\lambda = \sqrt{\frac{D_{Ca}}{k_{on}[B]}} \tag{2}$$

where $k_{on}$ is the on-rate for $Ca^{2+}$ binding ($k_{on} = 6 \times 10^8$ $M^{-1} s^{-1}$ for BAPTA and $1.5 \times 10^6$ $M^{-1} s^{-1}$ for EGTA) and the $[B]$ is the concentration of the buffer. The estimated unitary current of CRAC channels from noise analysis is ~3.5 fA at −80 mV in 22 mM $[Ca^{2+}]_o$ (*Zweifach and Lewis, 1993*). $i_{Ca}$ is likely about 40% larger due to the high open probability of single CRAC channels during brief (200–300ms) sweeps (*Prakriya and Lewis, 2006*), which would be expected to depress current noise.

The predicted $[Ca^{2+}]_i$ profiles in BAPTA and EGTA as a function of $d$ using an estimated $i_{Ca}$ value of 5 fA are shown in *Figure 5E*. The plots indicate that $[Ca^{2+}]_i$ is substantively reduced in 8 mM BAPTA compared to 10 mM EGTA. At less hyperpolarized membrane potentials (−60 to −80 mV), the local $[Ca^{2+}]$ would be expected to be smaller due to reduction in $i_{Ca}$. With the assumption that the number of channels and channel $P_o$ is unchanged by hyperpolarizing steps, we calculated $i_{Ca}$ at each potential by scaling the unitary current estimate (5 fA at −100 mV) by the ratio of the peak current at each potential to the current at −100 mV. At a distance of 3 nm (estimated to be the distance of the inactivation binding site from the CRAC channel pore *Zweifach and Lewis, 1995*) and using the scaled estimates of $i_{Ca}$, we predict that $[Ca^{2+}]_i$ is ~1.3 μM at a membrane voltage of −60 mV and 8 mM BAPTA.

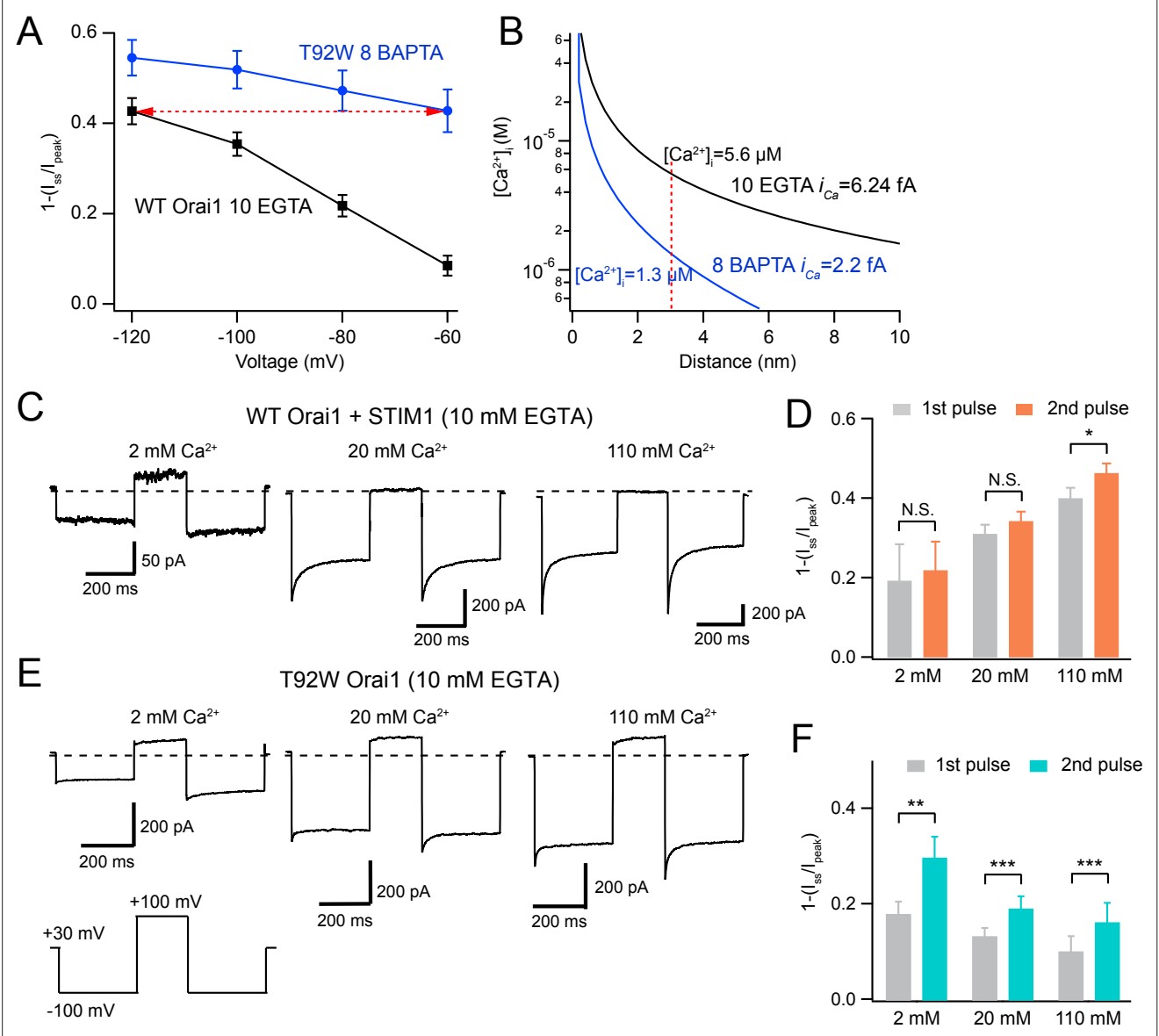

**Figure 6.** Depolarizing the cell promotes recovery from CDI in T92W Orai1 channels. (**A**) Comparison of extent of CDI in constitutively active T92W Orai1 (in 8 mM internal BAPTA) and WT Orai1 +STIM1 currents (in 10 mM internal EGTA). The extent of CDI at –60 mV in T92W is identical to CDI in WT Orai1 channels at –120 mV (dashed red arrow). The standard external solution contained 20 mM Ca²⁺ (see Methods). (**B**) $[Ca^{2+}]_i$ profiles in the two conditions that elicit similar levels of CDI in the experimental conditions in $A$. The unitary Ca²⁺ current at each potential was estimated by scaling the estimated $i_{Ca}$ at –100 mV with the ratio of the peak whole-cell current at the different voltages (see Methods). The profiles indicate that although CDI is similar, the $[Ca^{2+}]_i$ mediating CDI in T92W Orai1 at –60 mV is substantively lower than the $[Ca^{2+}]_i$ in WT Orai1 at –120 mV. (**C–F**) Analysis of recovery from inactivation in response to termination of Ca²⁺ influx with depolarizing voltage steps. (**C**) After a –100 mV hyperpolarizing voltage step to induce CDI, the cell was depolarized to +100 mV to abruptly terminate Ca²⁺ influx and lower submembrane $[Ca^{2+}]_i$. Following the recovery interval at +100 mV (200 ms), a second pulse to –100 mV was applied to evoke CRAC current. In WT Orai1 channels, the second pulse evokes Orai1 current with similar amplitude and inactivation as the first pulse. Holding potential is +30 mV. (**E**) Recovery from inactivation in T92W Orai1. Following a recovery step to +100 mV to terminate Ca²⁺ influx, a second hyperpolarizing pulse to –100 mV reveals significantly larger peak current and re-appearance of CDI indicating that abruptly terminating Ca²⁺ influx partly restores inactivation of T92W Orai1 channels in 10 mM EGTA. (**D and F**) The extent of inactivation ($1 − I_{ss}/I_{peak}$) in the first and second pulses at different extracellular Ca²⁺ concentrations. For T92W channels, CDI was enhanced by pre-pulse with +100 mV in 2 mM, 20 mM, and 110 mM external Ca²⁺ solutions. N=6–8 cells. Values are mean ± S.E.M. *:p<0.05, **p<0.01, ***: p<0.001 by paired t-test.

The online version of this article includes the following source data and figure supplement(s) for figure 6:

**Source data 1.** Paired pulse experiments reveals recovery of T92W inactivation through depolarizing steps.

**Figure supplement 1.** Recovery of inactivation of T92W and L138F Orai1 channels is promoted by depolarizing steps to +100 mV.

By contrast, in 10 mM EGTA and at –120 mV, local $[Ca^{2+}]_i$ is ~5.6 µM (*Figure 6B*). Thus, the similarity of inactivation of T92W in 8 mM BAPTA at –60 mV to that of WT Orai1 in 10 mM EGTA at –120 mV (*Figure 6A*) indicates that the $Ca^{2+}$-sensitivity of inactivation of T92W Orai1 is substantively increased, such that constitutively active T92W currents inactivate at *lower* concentrations of intracellular $Ca^{2+}$ compared to WT Orai1 channels (*Figure 6B*). Note that this conclusion is not dependent on the exact values of $i_{Ca}$ or the precise distance of the $Ca^{2+}$ binding site from the pore, for while altering these parameters would be expected to change the absolute values of $[Ca^{2+}]$ at the putative $Ca^{2+}$ binding site, the greater inactivation of T92W channels in BAPTA containing solutions still indicates that these channels show CDI at lower levels of $[Ca^{2+}]_i$ than WT Orai1 channels. An increase in $Ca^{2+}$ sensitivity of CDI in T92W Orai1 predicts that the mutant channel will enter the inactivated state more readily (and faster) than WT Orai1 (also see Discussion for a conceptual explanation of this phenomenon). As a result, in EGTA-containing solutions, steady-state inactivation of mutant channels at the holding potential is expected to be greater due to the higher local $[Ca^{2+}]_i$. Accordingly, hyperpolarizing steps are unable to elicit robust inactivation in the mutant channels in EGTA, likely explaining reduced CDI in this buffer.

## CDI of T92W Orai1 in EGTA is unmasked by rapid termination of $Ca^{2+}$ influx

If CDI of T92W Orai1 in EGTA is reduced relative to BAPTA solutions because channels are already at equilibrium with inactivation due to higher submembrane $[Ca^{2+}]_i$ at the holding potential (+30 mV), then abruptly lowering submembrane $[Ca^{2+}]_i$ should promote recovery of channels from inactivated states. We examined this idea using a two-pulse protocol in which we delivered a +100 mV depolarizing recovery pulse in between two hyperpolarizing pulses to –100 mV where we assessed CDI. In native CRAC channels of T-cells, it was previously shown that recovery from CDI occurs with a time course over tens of milliseconds, with ~90% recovery occurring in 200 ms at a recovery potential of –12 mV (*Zweifach and Lewis, 1995*). We reasoned that the depolarizing step to +100 mV should abruptly terminate $Ca^{2+}$ influx, and because submembrane $Ca^{2+}$ should be chelated by EGTA during the recovery pulse, recovery of T92W Orai1 channels from CDI will be promoted. As a consequence, a second pulse to –100 mV should manifest more inactivation. We tested this hypothesis at three extracellular $Ca^{2+}$ concentrations (2 mM, 20 mM, and 110 mM) applied to the same cells in the presence of 10 mM intracellular EGTA or 8 mM BAPTA in a paired-pulse protocol with a 200 ms step to +100 mV to terminate $Ca^{2+}$ influx and promote recovery of Orai1 from CDI. (*Figure 6C–F*).

Depolarizing the membrane potential to +100 mV in 2 mM extracellular $Ca^{2+}$ resulted in strong enhancement of the inward current in the second pulse (*Figure 6E and F*). Elevating $[Ca^{2+}]_o$ to 20 mM elicited smaller current enhancement of the second pulse relative to the first pulse, and raising $[Ca^{2+}]_o$ to 110 mM elicited even less recovery (*Figure 6E and F*). This result suggests that depolarizing pulses to +100 mV lower $[Ca^{2+}]_i$ and prevents the entry of channels into inactivated states, thereby allowing channels to recover from inactivation, and therefore, showing greater CDI during the second –100 mV pulse (also see scheme in Discussion). The extent of current recovery was, however, dependent on the extracellular $Ca^{2+}$ concentration with higher $Ca^{2+}$ concentrations causing less current recovery, as would be expected for a process involving $Ca^{2+}$-dependent accumulation of inactivation. These results indicate that T92 Orai1 channels are significantly inactivated in 10 mM EGTA at the holding potential and that recovery from inactivation is promoted by the depolarizing step to +100 mV which is predicted to lower submembrane $[Ca^{2+}]_i$. In 8 mM BAPTA, recovery is also promoted by depolarizing pulses to +100 mV (*Figure 6—figure supplement 1A and B*), although the degree recovery is reduced relative to EGTA. This is expected as substantial CDI already occurs in BAPTA indicating that at this level of buffering, resting inactivation at the holding potential (+30 mV) is less than in EGTA. As a result, there is diminished need for the membrane potential to be further depolarized to promote recovery from CDI. Recovery of L138F also resembled WT Orai1 rather than T92W Orai1 in EGTA solutions (*Figure 6—figure supplement 1C and D*). We think this is due to the very small current density of L138F Orai1, which is predicted to cause much lower levels of submembrane $[Ca^{2+}]_i$ elevations (relative to T92W Orai1) and hence reduced accumulation of channels into the inactivated state. Taken together, these observations are consistent with the interpretation that under weak buffering conditions, the intracellular $Ca^{2+}$ dependence of T92W Orai1 channels is strongly sensitized relative to WT channels, causing accumulation of

channels into inactivated state at the holding potential (+30 mV) thereby reducing further inactivation during hyperpolarizing steps.

## C- and N-terminal Orai1 mutations differentially affect T92W Orai1 inactivation

We next turned our attention to the molecular determinants of CDI of T92W channels. Previous studies have implicated several domains of the CRAC channel for CDI, including the ID region of STIM1, the Orai1 N-terminus and the Orai2/3 C-terminus (*Lee et al., 2009*; *Mullins et al., 2009*; *Mullins and Lewis, 2016a*). To determine the role of the Orai1 N- and C-termini for T92W CDI, we tested the effects of mutations in these domains on T92W Orai1 CDI. A previous study employing domain swap and site-directed mutations in Orai2 and Orai3 implicated the C-terminus as a key locus regulating Orai2/3 CDI (*Lee et al., 2009*). To address a potential role of the Orai1 C-terminus for T92W Orai1 inactivation, we therefore truncated the Orai1 C-terminus (*Figure 7A*). We found that deletion of the Orai1 C-terminus (Δ267–301) had no effect the constitutive activity of Orai1 T92W currents (*Figure 7B*), indicating that the Orai1 C-terminus is not necessary for the constitutive gating of this open mutant and reaffirming its STIM-independence for manifesting channel activity. However,

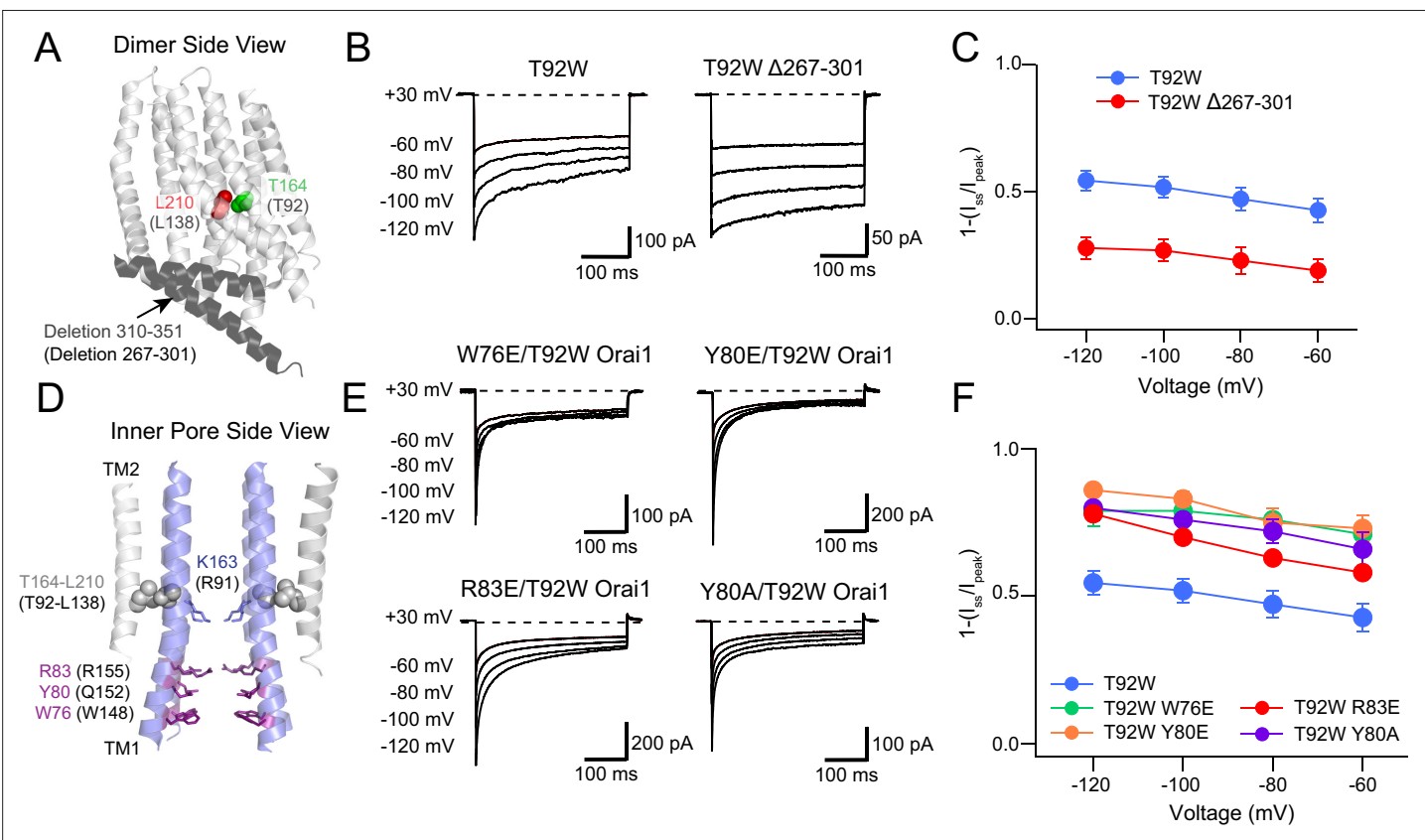

**Figure 7.** The Orai1 C-terminus contributes to T92W Orai1 CDI. (**A**) Side view of dOrai dimer (PDB ID: 4HKR; hOrai1 numbering in parentheses) showing location of T92 and L138 residues relative to the deleted C-terminal residues 267–301. The Orai1 C-terminus is shown in dark grey. (**B–C**) Deletion of the Orai1 C-terminus residues attenuates inactivation of T92W Orai1. Example traces showing CDI of T92W Orai1 and T92W Δ267–301 Orai1 mutant (**B**) The extent of inactivation is summarized in panel C. (**D**) Location of N-terminal residues (**W76, Y80 and R83**) previously implicated in regulating CDI in Orai1 channels. (**E**) Mutations of W76, Y80 and R83 in T92W Orai1 all strongly accelerated CDI compared to T92W Orai1 alone. (**F**) The extent of CDI plotted against membrane voltage in the indicated mutants. The inactivation quantification plot for T92W in *C* and *F* is the same as the data for this mutant in *Figure 4E*. N=4–17 cells. Values are mean ± S.E.M. The intracellular solution used was the standard internal solution with 8 mM BAPTA as the $Ca^{2+}$ buffer and the standard external solution contained 20 mM $Ca^{2+}$ (see Materials and methods).

The online version of this article includes the following source data and figure supplement(s) for figure 7:

**Source data 1.** Effects of N- and C-terminal mutations on T92W CDI.

**Figure supplement 1.** Deletion of the Orai1 N-terminus or introduction of a K85E mutation abrogates the constitutive T92W Orai1 current.

T92W Δ267–301 Orai1 channels showed markedly reduced inactivation (*Figure 7B and C*) with the extent of inactivation decreasing from ~50% to~25% at –100 mV. This finding indicates that the Orai1 C-terminus contributes to CDI of T92W Orai1. The presence of residual inactivation (20–30%) in the T92W Δ267–301 mutant however, implies that additional regions outside of the Orai1 C-terminus also make contributions to CDI.

At the Orai1 N-terminus, several reports have described changes in CDI in N-terminal Orai1 mutants (*Bergsmann et al., 2011*; *Mullins et al., 2016b*; *Zhang et al., 2019*). We found that truncation of the N-terminus (Δ2–85) abrogated T92W Orai1 currents (*Figure 7—figure supplement 1A and C*). Likewise, insertion of a K85E mutation in the N-terminus, which has been shown to abolish activity of both WT Orai1 channels gated by STIM1 as well as activity of many constitutively active Orai1 mutant channels, also abolished T92W channel activity (*Figure 7—figure supplement 1B and C*). The loss of channel activity in the Δ2–85 T92W and K85E/T92W mutants is similar to the loss of gating evoked by truncation of the N-terminus in STIM1-gated WT Orai1 and other constitutively active mutants (*Lis et al., 2010*; *McNally et al., 2013*; *Yeung et al., 2018*; *Tiffner et al., 2021*), indicating a generalized requirement of the N-terminus and the K85 residue for gating of WT and all constitutively active Orai1 mutant channels.

Mullins et al. have shown that mutation of the aromatic residues, Y80 and W76, in the N-terminus can enhance or abrogate CDI in Orai1 channels (*Mullins and Lewis, 2016a*; *Mullins et al., 2016b*). That study also implicated three positively charged residues (R91, K87, and R83) located in the inner pore (*Mullins et al., 2016b*) for CDI and it was postulated that the three residues, along with Y80 and W76 may form the inactivation gate, or are at least involved in the conformational changes mediating CDI (*Mullins and Lewis, 2016a*; *Mullins et al., 2016b*). To examine the role of these residues for T92W inactivation, we mutated Y80, R83, and W76 and studied whether these mutations similarly affected T92W CDI. The Y80A mutation was previously shown to strongly enhance the rate and extent of CDI of WT Orai1 channels activated by STIM1 (*Mullins et al., 2016b*). In a similar fashion, we found that constitutively active Y80A/T92W Orai1 currents inactivated markedly faster and to a greater degree compared to T92W Orai1 channels (*Figure 7E and F*). Surprisingly, however, the Y80E mutation, which was previously shown to abrogate Orai1 CDI, also enhanced the rate and extent of T92W inactivation (*Figure 7E and F*). Similarly, both the W76E and the R83E mutations, which were previously shown to eliminate CDI of WT Orai1 (*Mullins et al., 2016b*), accelerated the rate and increased the extent of CDI of T92W Orai1 currents (*Figure 7E and F*). These results are consistent with previous models indicating that the Orai1 N-terminus has an important role in inactivation gating. However, the enhancement of T92W Orai1 inactivation gating by mutations (Y80E, R83E, W76E) that were previously shown to abrogate CDI of WT Orai1 channels reveals a degree of complexity in the role of these residue in controlling inactivation and argue that the aromatics Y80 and W76 regulate inactivation of Orai1 channels through a mechanism distinct from serving as the inactivation gate.

## STIM1 normalizes the inactivation of T92W Orai1 channels

STIM1 is required for CDI of CRAC channels and is postulated to promote inactivation via functional coupling between the ID region of STIM1 with the Orai1 N-terminus (*Mullins and Lewis, 2016a*). What effect, if any, does STIM1 have on the intrinsic CDI of constitutively active T92W Orai1 channels? We examined this question by co-expressing STIM1 together with T92W Orai1 at a STIM/Orai1 cDNA ratio of 5:1, which is expected to be sufficient for STIM1-mediated inactivation of WT Orai1 channels. To our surprise, in the presence of STIM1, the EGTA/BAPTA buffer-dependence of T92W Orai1 CDI was reversed. T92W Orai1 currents in STIM1 showed robust CDI in the presence of 10 mM EGTA, which was comparable in extent and rate to WT Orai1 channels gated by STIM1 (*Figure 8C and D*). Conversely, replacing the intracellular buffer with 8 mM BAPTA (high buffering) strongly reduced CDI of T92W Orai1 channels (*Figure 8C and E*). As summarized in the steady-state inactivation plots, the behavior of T92W Orai1 channels co-expressing STIM1 was essentially similar to that of WT Orai1 channels activated by STIM1 (*Figure 8D and E*), and fully reversed from the behavior of STIM1-free T92W Orai1 channels. These results indicate that STIM1 'normalizes' the aberrant buffer dependence of inactivation of the constitutively active T92W Orai1 channels. Likewise, co-expressing STIM1 with constitutively active Y80E/T92W Orai1 mutant, which exhibits faster inactivation than T92W Orai1 single mutant in BAPTA containing solutions, also markedly decreased the rate and extent of CDI of this mutant relative to cells without STIM1 co-expression (*Figure 8—figure supplement 1*). These

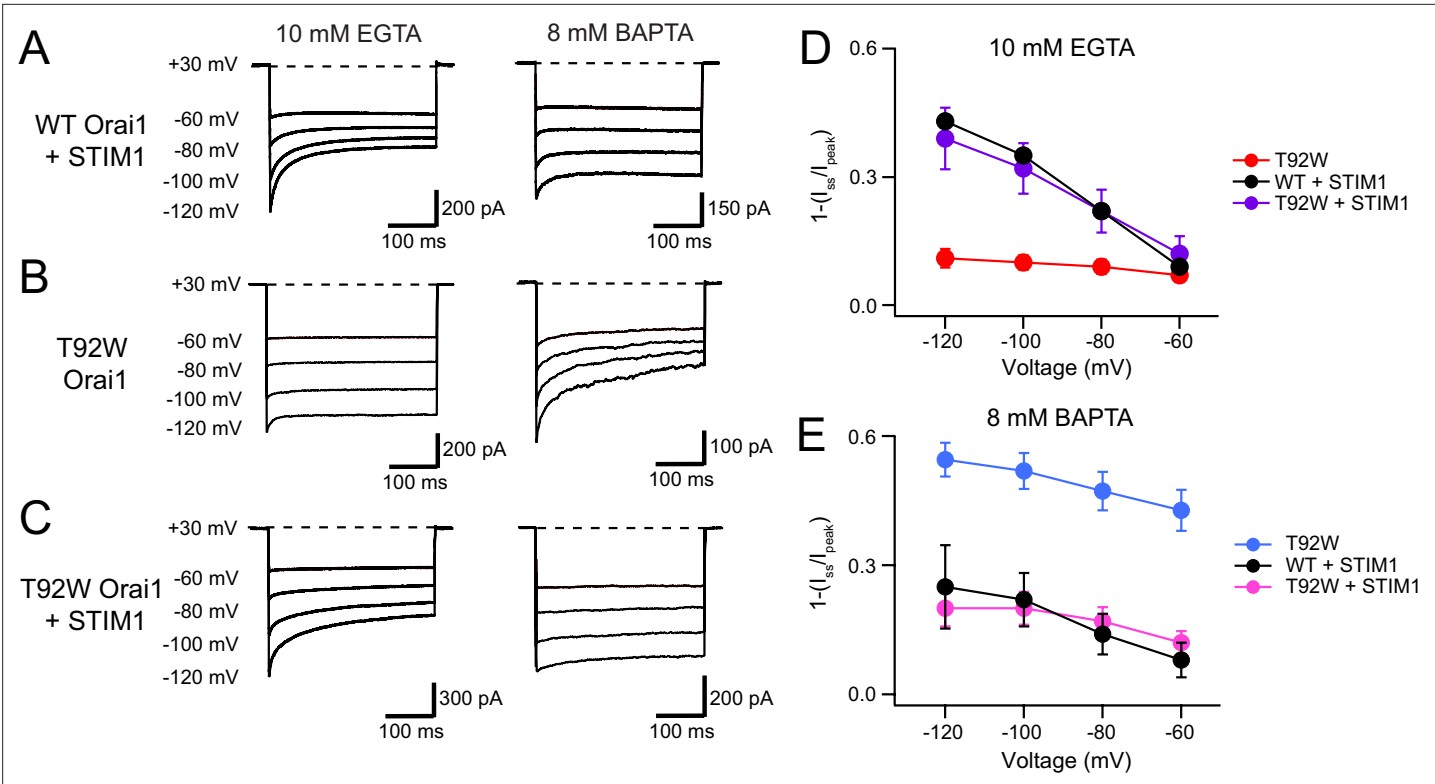

**Figure 8.** STIM1 modulates the Ca²⁺ sensitivity of T92W Orai1 CDI. (**A–B**) Representative traces of CDI in WT Orai1 +STIM1 (**A**) and T92W Orai1 (**B**) currents in the presence of EGTA (10 mM) or BAPTA (8 mM). (**C**) The addition of STIM1 normalizes the aberrant intracellular buffer dependence of T92W Orai1 CDI. In the presence of STIM1, CDI of T92 Orai1 is strongly enhanced in 10 mM EGTA compared to T92W currents in the absence of STIM1. Conversely, CDI in STIM1-expressing T92W Orai1 cells in 8 mM BAPTA is reduced relative to cells expressing T92W Orai1 alone (compare right traces in T92W in panels B and C), analogous to the behavior seen in WT Orai1 (**A**). (**D–E**) Summary plots showing the extent of CDI in the indicated conditions. The inactivation quantification plots for WT and T92W Orai1 are also shown in *Figure 5C–D*. All recordings were conducted in the standard external solution containing 20 mM Ca²⁺. N=4–17 cells. Values are mean ± S.E.M.

The online version of this article includes the following source data and figure supplement(s) for figure 8:

**Source data 1.** STIM1 restores the Ca²⁺ sensitivity of T92W CDI to that of WT Orai1.

**Figure supplement 1.** STIM1 modulates CDI of Y80E/T92W Orai1.

findings indicate that STIM1 modulates CDI of T92W Orai1 channels such that its Ca²⁺ sensitivity is similar to that of WT Orai1 channels gated by STIM1. The normalization of CDI by STIM1 is reminiscent of the 'normalization' of Ca²⁺ selectivity of V102C and other less Ca²⁺-selective constitutively active Orai1 mutants by STIM1 (*McNally et al., 2012*) and reaffirms the viewpoint that STIM1 plays an essential role for multiple aspects of Orai1 gating including activation, permeation, and inactivation.

## Discussion

A previous report showed that human mutation Orai1 L138F causes tubular aggregate myopathy with hypocalcemia in human patients (*Endo et al., 2015*). This syndrome is driven by dysregulated Ca²⁺ signaling in muscle cells secondary to constitutive Ca²⁺ entry through open Orai1 channels (*Endo et al., 2015*). In this study, we sought to understand the underlying mechanism of this pathogenic mutation and determined that steric clash between L138F and T92 on TM1 drives constitutive Orai1 activation. Large amino acid substitutions at either L138 or T92 that would increase the amount of contact between these two residues cause GOF Orai1 channels by disrupting the closed state of the pore. By contrast, mutations that reduce the packing density at this interface such as small or flexible substitutions at L138 lead to LOF Orai1 channels. These phenotypes point to a model wherein the L138-T92 nexus acts as a lever/pivot point at the TM2-TM1 interface to mediate channel activation. Unusually, constitutively active currents arising from L138F and T92W mutations show fast CDI with

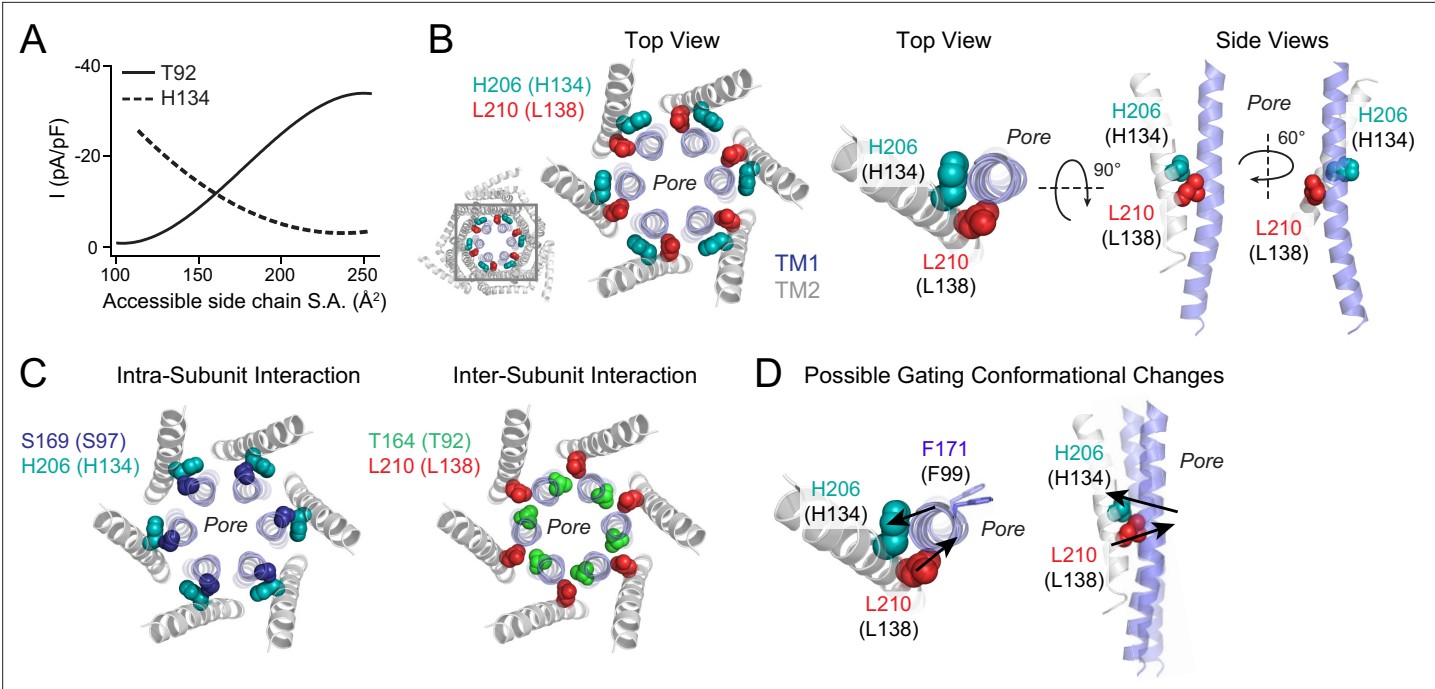

**Figure 9.** Schematic of the 'brace' formed by the H134 and L138 residues around TM1. (**A**) Polynomial fits of T92 and H134 mutant channel activity plotted against side chain surface area shows opposite dependence of these two positions on size. For both fits, charged residues (D/E/K/R) and G/P were excluded from this analysis because of their propensity to break α-helical secondary structure. (**B**) Left: Top view of the dOrai hexameric channel with the relative positions at the TM1-TM2/3 ring interface with H134 (teal) and L138 (red) shown. Middle and right: Top and side views of residues H134 (teal spheres) and L138 (red spheres) at the TM1 (blue) and TM2 (grey) interface. The two amino acids are on either side of the pore helix, approximately one turn apart. One TM2 and TM1 helix from each subunit are displayed for simplicity. (**C**) Intra-subunit interaction of S97 (navy) with H134 (teal) and inter-subunit interaction of T92 (green) with L138 (red) are shown to highlight the different interfaces of these pairs of interactions. (**D**) Proposed gating conformational change of L138 and H134 associated with channel opening. H134 and L138 are positioned one helical turn away from each other on opposite sides of the TM1 pore helix. Amino acid substitutions at L138 that cause TM1 to be pushed inwards, or H134 to be drawn outwards, cause channel activation via TM1 pore helix twisting and pore dilation to open the channel gate.

kinetics comparable to WT Orai1 channels gated by STIM1. CDI of T92W Orai1 shows increased $Ca^{2+}$ sensitivity and is not buffered by BAPTA, a chelator that reduces CDI of WT Orai1 channels. However, STIM1 co-expression normalizes the $Ca^{2+}$ sensitivity of T92W Orai1 to that of WT channels gated by STIM1. These results have important implications for the CDI mechanism and the role of STIM1 in the feedback inhibition process.

## The T92-L138 motif regulates Orai1 activation

The finding that the L138-T92 motif regulates Orai1 gating is notable as this locus is located very close to the H134 residue that functions as a steric "brake" at the interface between helices TM1, TM2, and TM3 to control Orai gating (*Frischauf et al., 2017*; *Yeung et al., 2018*). However, the side-chain size dependence of Orai1 activity at the L138-T92 locus follows an inverse pattern of what is observed for the H134 locus which is located one helical turn above on TM2 (*Figure 9A*). Introduction of bulky amino acids at T92 or L138 yield GOF channels, whereas small or flexible residues lead to LOF channels. At H134, by contrast, exactly the opposite is true: at this position small or flexible substitutions cause constitutive activity and large amino acids impede STIM1-mediated gating (*Yeung et al., 2018*).

What insights do the phenotypes of these mutants offer about the Orai1 channel activation process? The opposite dependencies of channel activity on the size of the side chain at H134 and L138 implies that these two residues are complementary in their effects. Both residues are on TM2, which is tilted diagonally relative to the membrane and TM1 pore helix. H134 and L138 are positioned squarely at the crossover between the two helices, and when viewed from the top, their side chains are in contact with opposite sides of TM1 (*Figure 9B*). We postulate that their complementary functional roles is related to several key differences in their positioning – (*i*) when viewed from the top, the

side-chains of H134 and L138 are on opposite sides of TM1, (*ii*) L138 is located one turn below H134, and (*iii*) the S97-H134 interaction is intra-subunit (*Yeung et al., 2018*) while T92-L138 interaction is inter-subunit (*Figure 9B and C*). We suggest that H134 and L138 function as pivot points for a 'brace' that stabilizes the pore helix. In this scenario, introducing steric pressure on TM1 via large amino acids substitutions at L138 or creating steric void at H134 outward through small substitutions both lead to channel activation, while 'pulling' at L138 or 'pushing' inward at H134 abolishes channel function. Although speculative, this pattern also raises the possibility that during STIM1-mediated gating, the TM1 helix is pushed outward by the lower portion of TM2 where L138 is positioned, causing it to tilt outward towards H134, leading to pore dilation in the hydrophobic stretch of the pore encompassing F99 that functions as the channel gate (*Yamashita et al., 2017*; *Figure 9D*). Moreover, because L138 and H134 are on different sides of TM1, they are also suitably positioned to help mediate rotation of the pore helix that accompanies pore dilation and channel opening (*Yamashita et al., 2017*; *Yeung et al., 2018*; *Figure 9D*).

A recent MD simulation study (*Zhang et al., 2021*) that examined the dOrai L210F mutation (equivalent to the human L138F Orai1 mutation) lends partial support to such a model. This study found that the introduced Phe side-chain at L210 is stabilized in a clockwise rotated state compared to WT channels (*Zhang et al., 2021*). The study also determined the conformational deflection of the L138F side chain was associated with opening of the hydrophobic gate in the pore through counterclockwise rotation of TM1 to increase pore waters as previously proposed for channel activation (*Yamashita et al., 2017*; *Yeung et al., 2018*; *Bulla et al., 2019*) and concurrent dilation of the inner pore (*Frischauf et al., 2017*; *Liu et al., 2019*; *Hou et al., 2020*). Together, the results of this study are compatible with our findings here that L138-TM1 interactions strongly influence the conformation(s) of the channel gate and suggest that a clockwise rotation of L138F as suggested by Zhang et al. would increases pore waters via a wetting transition to open the hydrophobic channel gate.

## The T92-L138 motif regulates fast CDI: implications for the CDI mechanism

Fast CDI is a distinguishing feature of CRAC channels and was first described in early recordings of CRAC currents in mast cells and T cells (*Hoth and Penner, 1993*; *Zweifach and Lewis, 1995*). Although several reports have described effects of mutations in different regions of the CRAC channel on CDI, a broader molecular understanding of fast CDI remains elusive. Key aspects of CDI that remain obscure include the identity of the $Ca^{2+}$ sensor for CDI, the identity and nature of the inactivation gate that closes the pore, and the coupling between the $Ca^{2+}$ sensor and the inactivation gate. A potential role for calmodulin (*Mullins et al., 2009*) and STIM1 (*Derler et al., 2009*; *Scrimgeour et al., 2009*; *Mullins and Lewis, 2016a*) as the $Ca^{2+}$ sensors for CDI were suggested in earlier studies, but subsequent evidence raised questions about their role (*Mullins and Lewis, 2016a*; *Mullins et al., 2016b*), leaving the identity of the $Ca^{2+}$ sensor a mystery. Nevertheless, a general consensus that STIM1 is necessary for CDI has emerged, and in particular, the ID region of STIM1 (encompassing residues 474–491) containing several acidic residues is implicated as an essential domain for CDI. STIM1 mutants lacking $ID_{STIM1}$ fail to show CDI, and mutating specific acidic residues in this region accelerate, or in some cases, diminish CDI (*Mullins and Lewis, 2016a*; *Mullins et al., 2016b*). Within Orai1, mutational analysis and domain swap experiments between Orai isoforms have implicated the cytosolic domains, in particular the Orai1 N- and the C-termini (*Lee et al., 2009*; *Srikanth et al., 2010*; *Frischauf et al., 2011*; *Mullins et al., 2016b*) in CDI. From these studies, the most generally accepted view is that the $ID_{STIM1}$ allosterically interacts with the Orai1 inner pore to mediate CDI (*Mullins and Lewis, 2016a*; *Mullins et al., 2016b*). The aromatics Y80 and W76 and to a lesser extent, three positively charged residues (R91, K87, and R83), have been strongly implicated (*Mullins et al., 2016b*) raising the possibility that this region of the inner pore may function as the channel gate for CDI.

Against this backdrop, the finding that Orai1 L138F and T92W channels show rapid CDI in the absence of STIM1 provide several new insights on the mechanisms of CDI. First, this result indicates that STIM1 is not essential for mediating CDI, nor does it function as the $Ca^{2+}$ sensor. Rather, the results suggest that the $Ca^{2+}$ sensor is likely located within the Orai1 protein itself, or a closely associated accessory subunit. We favor the idea that the $Ca^{2+}$ binding site is located on Orai1 itself and our early experiments suggest that the Orai1 C-terminus likely has a key role in this process. Second, we find that CDI of T92W Orai1 is much more prominent in BAPTA-containing internal solutions

compared to EGTA-based solutions. This is the exact reverse of what is seen for WT Orai1 channels. Our analysis suggests that this is due to enhanced $Ca^{2+}$ sensitivity of T92W Orai1 channels for CDI compared to WT Orai1 channels, which is normalized by STIM1. Thus, STIM1 functions to tune the $Ca^{2+}$-sensitivity of CDI.

The implications of the increased $Ca^{2+}$-sensitivity of inactivation in T92W Orai1 channels for macroscopic Orai1 currents can be analyzed in terms of the simple reaction scheme:

$$Ca \; + \; O \underset{k2}{\overset{k1}{\rightleftharpoons}} Ca \; - \; O \underset{\beta}{\overset{\alpha}{\rightleftharpoons}} I \tag{3}$$

where $Ca^{2+}$ ions bind to open ($O$) CRAC channels with forward and reverse rate constants of $k_1$ and $k_2$ and drive them into the inactivated state ($I$) with forward and reverse rate constants of and $\beta$ respectively. Purely for simplicity, we have assumed first-order kinetics for the binding and gating steps, although the actual state diagram is likely more complex (as already hinted by the presence of at least two exponentials for the inactivation time course). Nevertheless, the simple scheme above is instructive in illuminating the potential mechanisms of the change in CDI in T92W Orai1 channels. The analysis of *Figure 6A* suggests that the $Ca^{2+}$ sensitivity of T92W Orai1 for CDI is strongly enhanced compared to WT Orai1 channels (i.e. $K_d = k_2/k_1$ is reduced in T92W Orai1). This change in $K_d$ predicts that at a given $[Ca^{2+}]_i$, the forward reaction will be more strongly favored in the mutant compared to WT Orai1, and the overall reaction is predicted to reach equilibrium at lower $Ca^{2+}$ concentrations in T92W compared to WT Orai1 channels. In fact, with weak $Ca^{2+}$ buffering (EGTA or low concentrations of BAPTA) and recurring (every 1 s) steps to –100 mV, T92W inactivation could reach equilibrium at the holding potential (+30 mV) itself such that hyperpolarizing steps are unable to cause additional inactivation. As a result, membrane hyperpolarization fails to elicit additional inactivation in the presence of EGTA.

Although T92W Orai1 channels show enhanced $Ca^{2+}$ sensitivity for CDI, co-expression of STIM1 'normalizes' the $Ca^{2+}$-dependence of the mutant channels (*Figure 8*). The 'normalization' of CDI $Ca^{2+}$ sensitivity is reminiscent of the normalization of ion selectivity of poorly $Ca^{2+}$ selective V102C/A Orai1 channels by STIM1 (*McNally et al., 2012*; *Derler et al., 2013*). A change in $Ca^{2+}$ sensitivity of CDI could presumably occur via STIM1-driven change in the conformation of the C-terminus, which harbors several acidic residues that could bind $Ca^{2+}$ and which are located in close vicinity to residues L273 and L276 that are critical for STIM1 binding (*Muik et al., 2008*; *Navarro-Borelly et al., 2008*). Interaction of STIM1 with the Orai1 C-terminus could potentially alter $Ca^{2+}$ binding to this region to modulate CDI. Interestingly, truncation of the C-terminus does not completely eliminate CDI (*Figure 7B and C*) indicating the C-terminus is not the sole determinant of CDI. Other potential candidates that could function as the $Ca^{2+}$ sensors for CDI could include the Glu selectivity filter, which has been previously implicated in CDI (*Yamashita et al., 2007*) and the TM2-3 loop (*Srikanth et al., 2010*) which contains several acidic residues.

In the N-terminus, mutations (W76E, Y80E, R83E) that abrogate CDI of WT Orai1 channels gated by STIM1 (*Mullins and Lewis, 2016a*; *Mullins et al., 2016b*) substantially accelerated and increased CDI of T92W Orai1 channels. These phenotypes reinforce that notion that the inner pore is a key molecular determinant of CDI in Orai1 channels, but the acceleration of CDI indicates that this region may not function as the inactivation gate.

A key question is why mutations at the Orai1 L138-T92 locus even show CDI. This stands in sharp contrast to all other previously described GOF mutations (at V102, H134, P245 and others) which do not display CDI. We propose that the Orai1 L138-T92 locus mutations described here stabilize the inner pore of the channel in a conformation that is permissive for local $Ca^{2+}$ to drive inactivation, and this occurs in a manner similar to that induced by STIM1 itself. While the other GOF mutations activate the channel gate and allow ion conduction, the inner pore is not in the correct conformational state permissive for $Ca^{2+}$ to stimulate CDI. This interpretation would also imply that the molecular determinants of activation and inactivation differ in important ways and more studies are needed to resolve this major question.

The increased $Ca^{2+}$ sensitivity of T92W and L138F CDI likely also explains the high $I_{Na}/I_{Ca}$ current ratio observed for these mutants (*Figure 1*), as inactivation is relieved in $Na^+$-containing solutions. The presence of CDI in the L138F human mutation may also explain previous observations indicating that this pathological mutation raises resting cytoplasmic $[Ca^{2+}]$ to a lesser degree than other gain-of-function

human mutations (e.g. S97C, G98S) and evokes milder symptoms (tubular aggregate myopathy but not hypocalcemia) (*Endo et al., 2015*; *Garibaldi et al., 2016*; *Böhm et al., 2017*). The milder defect in the L138F mutant may be related to partial blunting of the steady-state open probability of the GOF channel thereby somewhat blunting constitutive $Ca^{2+}$ entry.

Taken together, this evidence indicates that CDI of Orai1 occurs independently of STIM1 with the $Ca^{2+}$ binding site likely located within Orai1 itself, possibly at the Orai1 C-terminus. The precise identity and structural basis of the $Ca^{2+}$ binding and how $Ca^{2+}$ binding to the sensor is communicated to the pore to evoke channel closure remains to be understood but the results here provide a way forward to address these questions in a simplified one-component system using a constitutively active Orai1 mutant that shows CDI in the absence of STIM1.

## Materials and methods

### Cells

HEK293-H cells were maintained in suspension at 37 °C with 5% $CO_2$ in CD293 medium supplemented with 4 mM GlutaMAX (Invitrogen). The HEK293 cell line is a permanent line established from primary embryonic human kidney and transformed with sheared human adenovirus type 5 DNA. The E1A adenovirus gene is expressed in these cells to optimize protein production. HEK293-H cells were cloned from the original 293 cell line and adapted to CD293 serum-free medium for growth in suspension. Cell line identity has been authenticated by Thermo- Fisher Scientific, and cells were tested negative for mycoplasma by qPCR detection assay. For imaging and electrophysiology, cells were plated onto poly-L-lysine coated coverslips one day before transfection and grown in a medium containing 44% DMEM (Corning), 44% Ham's F12 (Corning), 10% fetal bovine serum (HyClone), 2 mM glutamine, 50 U/ml penicillin and 50 µg/ml streptomycin.

### Plasmids and transfections

The Orai1 mutants employed for electrophysiology were engineered into a pEYFP-N1 vector (Clonetech) to produce C-terminally tagged Orai1-YFP proteins (*Navarro-Borelly et al., 2008*). mCherry-STIM1 and CFP-CAD were kind gifts of Dr. R. Lewis (Stanford University, USA). All mutants were generated by the QuikChange Mutagenesis Kit (Agilent Technologies) and the mutations were confirmed by DNA sequencing. For electrophysiology, the indicated Orai1 constructs were transfected into HEK293-H cells either alone (200 ng DNA per coverslip) or together with STIM1 (100 ng Orai1 and 500 ng STIM1 DNA per coverslip). For FRET microscopy experiments, cells were transfected with Orai1-YFP and CFP-CAD constructs (100 ng each per coverslip). All transfections were performed using Lipofectamine 2000 (Thermo Fisher Scientific) 24–48 hours prior to electrophysiology or imaging experiments.

### Solutions and chemicals

The standard 20 mM $Ca^{2+}$ extracellular Ringer's solution used for electrophysiological experiments contained 135 mM NaCl, 4.5 mM KCl, 20 mM $CaCl_2$, 1 mM $MgCl_2$, 10 mM D-glucose, and 5 mM HEPES (pH 7.4 with NaOH). 110 mM $Ca^{2+}$ solution contained 110 mM $CaCl_2$, 10 mM D-glucose, and 5 mM HEPES (pH 7.4 with NaOH). The divalent-free (DVF) solution contained 150 mM NaCl, 10 mM HEDTA, 1 mM EDTA, and 10 mM HEPES (pH 7.4 with NaOH). 10 mM TEA-Cl was added to prevent contamination from voltage-gated $K^+$ channels. All internal solutions contained 8 mM $MgCl_2$ and 10 mM HEPES (pH 7.2 with CsOH). The standard 8 mM BAPTA internal solution (which was used in the experiments shown in all Figures unless otherwise indicated) contained 135 mM Cs aspartate and 8 mM BAPTA. The 20 mM BAPTA solution contained 95 Cs asparatate and the 0.8 mM BAPTA solution contained 145 mM Cs aspartate (all pH 7.2). The 10 mM EGTA solution contained 130 mM Cs aspartate, and the 20 mM EGTA solution contained 110 mM Cs aspartate (pH 7.2).

### Electrophysiology

Currents were recorded in the standard whole-cell configuration at room temperature on an Axopatch 200B amplifier (Molecular Devices) interfaced to an ITC-18 input/output board (Instrutech). Routines developed by R. S. Lewis (Stanford) on the Igor Pro software (Wavemetrics) were employed for stimulation, data acquisition and analysis. Data are corrected for the liquid junction potential of the pipette

solution relative to Ringer's in the bath (–10 mV). The holding potential was +30 mV. The standard voltage stimulus consisted of a 100 ms step to –100 mV followed by a 100 ms ramp from –100 to +100 mV applied at 1 s intervals. For voltage families, steps to –120 mV, –100 mV, –80 mV, and –60 mV were 300 ms each. In the paired-pulse experiment, the holding potential was +30 mV and the two steps were to –100 mV for 300 ms each separated by a step to +100 mV for 200 ms in between the two hyperpolarizing steps. In experiments where Orai1 was co-expressed with STIM1, $I_{CRAC}$ was typically activated by passive depletion of ER $Ca^{2+}$ stores by intracellular dialysis of 8 mM BAPTA. All currents were acquired at 5 kHz and low pass filtered with a 1 kHz Bessel filter built into the amplifier. All data were corrected for leak currents collected in 100–200 µM $LaCl_3$.

## Data analysis

Analysis of current amplitudes was typically performed by measuring the peak currents during the –100 mV pulse. Specific mutants were categorized as gain-of-function if their currents exceeded 2 pA/pF, which is more than ten times the current density of WT Orai1 without STIM1. Reversal potentials were measured from the average of several leak-subtracted sweeps in each cell. For CDI, the extent of inactivation was determined from the relative decrease in current (relative to the peak current) during the voltage pulse and quantified as $(1-I_{ss}/I_{peak})$ where $I_{ss}$ is the current at the end of the 300 ms hyperpolarizing step and $I_{peak}$ is the peak current immediately following the hyperpolarizing step. The time course of CDI was fit with a double-exponential function and the fast and slow time constants ($\tau_{fast}$ and $\tau_{slow}$) were determined from the fits. All fitting was done using the built-in routines in Igor Pro v6.12. $[Ca^{2+}]_i$ profiles were calculated using *equations 1 and 2* using the parameters for $i_{Ca}$, $D_{Ca}$, and buffer concentration as indicated in the Results. The $i_{Ca}$ at –60 mV (2.2 fA) and –120 mV (~6.2 fA) was calculated by linearly scaling the unitary current at –100 mV (5 fA) with the altered driving force for $Ca^{2+}$ entry. All data are expressed as means ± SEM. For datasets with two groups, statistical analysis was performed with two-tailed t test to compare between control and test conditions. For datasets with greater than two groups, one-way ANOVA followed by Tukey post-hoc test was used to compare groups. Statistical analysis was performed with a confidence level of 95%, and results with p<0.05 were considered statistically significant. Significance is denoted as *p<0.05, **p<0.01, ***p<0.001.

## Atomic packing analysis

Atomic packing analysis was performed as in our previous study (*Yeung et al., 2018*). Briefly, it carried out using the programs REDUCE and PROBE that simulates rolling a 0.25 Å radius sphere along the van der Waals surfaces. Locations where the probe sphere contacts two surfaces are marked (with a 'dot') that classifies whether the surfaces are in wide contact, close contact, overlapped, or clashing. The resulting contact dot scores were summed for all atoms of each residue and displayed using PyMOL on a heat map that shows the degree of contacts.

## FRET microscopy

HEK293-H cells transfected with Orai1-YFP and CFP-CAD DNA constructs were imaged using wide-field epifluorescence microscopy on an IX71 inverted microscope (Olympus, Center Valley, PA). Cells were imaged with a 60 X oil immersion objective (UPlanApo NA 1.40), a 175 W Xenon arc lamp (Sutter, Novatao, CA), and excitation and emission filter wheels (Sutter, Novato, CA). At each time point, three sets of images (CFP, YFP, and FRET) were captured on a cooled EM-CCD camera (Hamamatsu, Bridge-water, NJ) using optical filters specific for the three images as previously described. Image acquisition and analysis was performed with SlideBook software (Imaging Innovations Inc, Denver, CO). Images were captured at exposures of 100–500 ms with 1X1 binning. Lamp output was attenuated to 25% by a 0.6 ND filter in the light path to minimize photobleaching. All experiments were performed at room temperature.

FRET analysis was performed as previously described (*Navarro-Borelly et al., 2008*). The microscope-specific bleed-through constants (a=0.12; b=0.008; c=0.002 and d=0.33) were determined from cells expressing cytosolic CFP or YFP alone. The apparent FRET efficiency was calculated from background-subtracted images using the formalism (*Zal and Gascoigne, 2004*):

$$E_{FRET} = \frac{F_c}{F_c + GI_{DD}}$$

where

$$F_c = I_{DA} \cdot aI_{AA} - dI_{DD}$$

$I_{DD}$, $I_{AA}$ and $I_{DA}$ refer to the background subtracted CFP, YFP, and FRET images, respectively. The instrument dependent $G$ factor had the value 1.85±0.1. E-FRET analysis was restricted to cells with YFP/CFP ratios in the range of 2–6 to ensure that E-FRET was compared across identical acceptor to donor ratios, and measurements were restricted to regions of interest drawn at the plasma membrane.

## Acknowledgements

We thank members of the laboratory and RS Lewis and CJ Lingle for helpful discussions. This work was supported by NIH grants R01 NS057499 and R01 NS115508 to MP. PS-WY was supported by NIH predoctoral fellowship F31NS101830.

## Additional information

### Competing interests

Murali Prakriya: Reviewing editor, *eLife*. The other authors declare that no competing interests exist.

### Funding

| Funder | Grant reference number | Author |
| --- | --- | --- |
| National Institutes of Health | R01 NS057499 | Murali Prakriya |
| National Institutes of Health | R01 NS115508 | Murali Prakriya |
| National Institutes of Health | F31NS101830 | Priscilla S-W Yeung |

The funders had no role in study design, data collection and interpretation, or the decision to submit the work for publication.

### Author contributions

Priscilla S-W Yeung, Conceptualization, Resources, Data curation, Formal analysis, Funding acquisition, Validation, Investigation, Visualization, Methodology, Writing - original draft, Project administration, Writing - review and editing; Megumi Yamashita, Conceptualization, Data curation, Formal analysis, Validation, Investigation, Visualization, Methodology, Writing - original draft, Writing - review and editing; Murali Prakriya, Conceptualization, Resources, Data curation, Formal analysis, Supervision, Funding acquisition, Validation, Investigation, Visualization, Methodology, Writing - original draft, Project administration, Writing - review and editing

### Author ORCIDs

Priscilla S-W Yeung http://orcid.org/0000-0001-5400-8639
Megumi Yamashita http://orcid.org/0000-0003-0196-0428
Murali Prakriya http://orcid.org/0000-0003-0781-4480

### Decision letter and Author response

Decision letter https://doi.org/10.7554/eLife.82281.sa1
Author response https://doi.org/10.7554/eLife.82281.sa2

## Additional files

### Supplementary files

• MDAR checklist

**Data availability**

Source data files containing the numerical data used in Figures 1–8 and the associated figure supplements have been provided.

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

# Appendix 1

### Appendix 1—key resources table

| Reagent type (species) or resource | Designation | Source or reference | Identifiers | Additional information |
|---|---|---|---|---|
| Cell line (*Homo-sapiens*) | HEK293-H | Thermo Fisher Scientific | 11631017 | RRID:CVCL_6643 |
| Commercial assay or kit | QuikChange II XL Site-Directed Mutagenesis Kit | Agilent | 200522 | |
| Transfected construct (human) | Orai1-YFP | Clontech | *Navarro-Borelly et al., 2008* | |
| Transfected construct (human) | mCherry-STIM1 | Richard Lewis (Stanford) | | |
| Transfected construct (human) | CFP-CAD | Richard Lewis (Stanford) | | |
| Chemical compound, drug | Lipofectamine 2000 | Thermo Fisher Scientific | 11668019 | |
| Chemical compound, drug | EGTA | Sigma Aldrich | E3889 | |
| Chemical compound, drug | BAPTA | Thermo Fisher Scientific | B1212 | |
| Sequence-based reagent | mutagenesis primers for Orai1 L138A | IDT | This paper | gcaggtgctgatcatggccgcaaacaggtgcaca tgtgcacctgtttgcggccatgatcagcacctgc |
| Sequence-based reagent | mutagenesis primers for Orai1 L138C | IDT | *Yeung et al., 2018* | tgcaggtgctgatcatgcacgcaaacaggtgcacag ctgtgcacctgtttgcgtgcatgatcagcacctgca |
| Sequence-based reagent | mutagenesis primers for Orai1 L138D | IDT | This paper | gcaggtgctgatcatgtccgcaaacaggtgcaca tgtgcacctgtttgcggacatgatcagcacctgc |
| Sequence-based reagent | mutagenesis primers for Orai1 L138E | IDT | This paper | gatgcaggtgctgatcatctccgcaaacaggtgcacagc gctgtgcacctgtttgcggagatgatcagcacctgcatc |
| Sequence-based reagent | mutagenesis primers for Orai1 L138F | IDT | This paper | caggtgctgatcatgaacgcaaacaggtgcaca tgtgcacctgtttgcgttcatgatcagcacctg |
| Sequence-based reagent | mutagenesis primers for Orai1 L138G | IDT | This paper | gcaggtgctgatcatgcccgcaaacaggtgcaca tgtgcacctgtttgcgggcatgatcagcacctgc |
| Sequence-based reagent | mutagenesis primers for Orai1 L138H | IDT | This paper | caggtgctgatcatgtgcgcaaacaggtgca tgcacctgtttgcgcacatgatcagcacctg |
| Sequence-based reagent | mutagenesis primers for Orai1 L138I | IDT | This paper | caggtgctgatcatgatcgcaaacaggtgcaca tgtgcacctgtttgcgatcatgatcagcacctg |
| Sequence-based reagent | mutagenesis primers for Orai1 L138K | IDT | This paper | gatgcaggtgctgatcatcttcgcaaacaggtgcacagc gctgtgcacctgtttgcgaagatgatcagcacctgcatc |
| Sequence-based reagent | mutagenesis primers for Orai1 L138M | IDT | This paper | atgcaggtgctgatcatcatcgcaaacaggtgcacag ctgtgcacctgtttgcgatgatgatcagcacctgcat |

*Appendix 1 Continued on next page*

*Appendix 1 Continued*

| Reagent type (species) or resource | Designation | Source or reference | Identifiers | Additional information |
|---|---|---|---|---|
| Sequence-based reagent | mutagenesis primers for Orai1 L138N | IDT | This paper | tgcaggtgctgatcatgttcgcaaacaggtgcacag ctgtgcacctgtttgcgaacatgatcagcacctgca |
| Sequence-based reagent | mutagenesis primers for Orai1 L138P | IDT | This paper | caggtgctgatcatgggcgcaaacaggtgca tgcacctgtttgcgcccatgatcagcacctg |
| Sequence-based reagent | mutagenesis primers for Orai1 L138Q | IDT | This paper | tgcaggtgctgatcatctgcgcaaacaggtgcac gtgcacctgtttgcgcagatgatcagcacctgca |
| Sequence-based reagent | mutagenesis primers for Orai1 L138R | IDT | This paper | caggtgctgatcatgcgcgcaaacaggtgca tgcacctgtttgcgcgcatgatcagcacctg |
| Sequence-based reagent | mutagenesis primers for Orai1 L138S | IDT | This paper | tgcaggtgctgatcatgctcgcaaacaggtgcacag ctgtgcacctgtttgcgagcatgatcagcacctgca |
| Sequence-based reagent | mutagenesis primers for Orai1 L138T | IDT | This paper | tgcaggtgctgatcatggtcgcaaacaggtgcacag ctgtgcacctgtttgcgaccatgatcagcacctgca |
| Sequence-based reagent | mutagenesis primers for Orai1 L138V | IDT | This paper | aggtgctgatcatgaccgcaaacaggtgcac gtgcacctgtttgcggtcatgatcagcacct |
| Sequence-based reagent | mutagenesis primers for Orai1 L138W | IDT | This paper | gatgcaggtgctgatcatccacgcaaacaggtgcacagc gctgtgcacctgtttgcgtggatgatcagcacctgcatc |
| Sequence-based reagent | mutagenesis primers for Orai1 L138Y | IDT | This paper | ggatgcaggtgctgatcatatacgcaaacaggtgcacagcc ggctgtgcacctgtttgcgtatatgatcagcacctgcatcc |
| Sequence-based reagent | mutagenesis primers for Orai1 T92A | IDT | This paper | cagagccgaggcccggctggagg cctccagccgggcctcggctctg |
| Sequence-based reagent | mutagenesis primers for Orai1 T92C | IDT | *Yeung et al., 2018* | gagcagagccgagcaccggctggaggct agcctccagccggtgctcggctctgctc |
| Sequence-based reagent | mutagenesis primers for Orai1 T92D | IDT | This paper | gagcagagccgagtcccggctggaggct agcctccagccgggactcggctctgctc |
| Sequence-based reagent | mutagenesis primers for Orai1 T92E | IDT | This paper | ggagagcagagccgactcccggctggaggcttt aaagcctccagccgggagtcggctctgctctcc |
| Sequence-based reagent | mutagenesis primers for Orai1 T92F | IDT | This paper | gagcagagccgagaaccggctggaggct agcctccagccggttctcggctctgctc |
| Sequence-based reagent | mutagenesis primers for Orai1 T92G | IDT | This paper | gagcagagccgagcccggctggaggct agcctccagccggggctcggctctgctc |
| Sequence-based reagent | mutagenesis primers for Orai1 T92H | IDT | This paper | gagcagagccgagtgccggctggaggct agcctccagccggcactcggctctgctc |
| Sequence-based reagent | mutagenesis primers for Orai1 T92I | IDT | This paper | agcagagccgagatccggctggagg cctccagccggatctcggctctgct |
| Sequence-based reagent | mutagenesis primers for Orai1 T92K | IDT | This paper | agcagagccgacttccggctggaggctttaagc gcttaaagcctccagccggaagtcggctctgct |
| Sequence-based reagent | mutagenesis primers for Orai1 T92L | IDT | This paper | cggagagcagagccgatagccggctggaggcttta taaagcctccagccggctatcggctctgctctccg |

*Appendix 1 Continued on next page*

*Appendix 1 Continued*

| Reagent type (species) or resource | Designation | Source or reference | Identifiers | Additional information |
|---|---|---|---|---|
| Sequence-based reagent | mutagenesis primers for Orai1 T92M | IDT | This paper | agcagagccgacatccggctggaggctttaagc gcttaaagcctccagccggatgtcggctctgct |
| Sequence-based reagent | mutagenesis primers for Orai1 T92N | IDT | This paper | agcagagccgagttccggctggagg cctccagccggaactcggctctgct |
| Sequence-based reagent | mutagenesis primers for Orai1 T92P | IDT | This paper | cagagccgagggccggctggagg cctccagccggccctcggctctg |
| Sequence-based reagent | mutagenesis primers for Orai1 T92Q | IDT | This paper | ggagagcagagccgactgccggctggaggcttt aaagcctccagccggcagtcggctctgctctcc |
| Sequence-based reagent | mutagenesis primers for Orai1 T92R | IDT | This paper | gcagagccgacctccggctggaggctttaa ttaaagcctccagccggaggtcggctctgc |
| Sequence-based reagent | mutagenesis primers for Orai1 T92S | IDT | This paper | gcagagccgagctccggctggag ctccagccggagctcggctctgc |
| Sequence-based reagent | mutagenesis primers for Orai1 T92V | IDT | This paper | gagcagagccgagacccggctggaggct agcctccagccgggtctcggctctgctc |
| Sequence-based reagent | mutagenesis primers for Orai1 T92W | IDT | This paper | ggagagcagagccgaccaccggctggaggcttt aaagcctccagccggtggtcggctctgctctcc |
| Sequence-based reagent | mutagenesis primers for Orai1 T92Y | IDT | This paper | cggagagcagagccgaataccggctggaggcttta taaagcctccagccggtattcggctctgctctccg |
| Sequence-based reagent | mutagenesis primers for Orai1 K85E | IDT | *McNally et al., 2013* | ggaggctttaagctcggcgcggctcaagt acttgagccgcgccgagcttaaagcctcc |
| Sequence-based reagent | mutagenesis primers for Orai1 W76E | IDT | *Mullins et al., 2016b* | caagtagagcttgcgctcggacagcgcctgcatg catgcaggcgctgtccgagcgcaagctctacttg |
| Sequence-based reagent | mutagenesis primers for Orai1 Y80A | IDT | *Mullins et al., 2016b* | ggcgcggctcaaggcgagcttgcgccag ctggcgcaagctcgccttgagccgcgcc |
| Sequence-based reagent | mutagenesis primers for Orai1 Y80E | IDT | *Mullins et al., 2016b* | tggcgcggctcaactcgagcttgcgccag ctggcgcaagctcgagttgagccgcgcca |
| Sequence-based reagent | mutagenesis primers for Orai1 R83E | IDT | *Mullins et al., 2016b* | ggaggctttaagcttggcctcgctcaagtagagcttgcg cgcaagctctacttgagcgaggccaagcttaaagcctcc |
| Sequence-based reagent | mutagenesis primers for Orai1 del267-301 | IDT | *Yamashita et al., 2007* | gcgtccagctgcacatccaccattgccac gtggcaatggtggatgtgcagctggacgc |
| Sequence-based reagent | mutagenesis primers for Orai1 del267-301 | IDT | This paper | ggcgaccggtggatcggtcttatggctaac gttagccataagaccgatccaccggtcgcc |

