## [Editor Report]

This manuscript provides strong evidence on the molecular basis of a mutation (L138F) in Orai1 channels that is associated with tubular aggregate myopathy. This disease-related mutation results in a gain of function of Orai1 channels due to a steric clash clash between TM1 and TM2. The study further suggests that Ca^2+^-dependent Inactivation (CDI) is an intrinsic feature of Orai1 channels and that STIM1 fine-tunes CDI.

---

## [Decision Letter]

**Decision letter after peer review:**

Thank you for submitting your article "A human tubular aggregate myopathy mutation unmasks STIM1-independent rapid inactivation of Orai1 channels" for consideration by *eLife*. Your article has been reviewed by 3 peer reviewers, one of whom is a member of our Board of Reviewing Editors, and the evaluation has been overseen by Richard Aldrich as the Senior Editor. The following individual involved in the review of your submission has agreed to reveal their identity: Peter B. Stathopulos (Reviewer #3).

Essential revisions:

1) Additional experiments are required to strengthen data related to CDI of the T92W Orai1 mutant. The authors propose that constitutive T92W Orai1 currents inactivate at a lower intracellular Ca^2+^ concentration compared to WT Orai1. Based on a 2-step CDI protocol from a holding membrane potential of +100mV and 10mM EGTA in the patch pipette, the authors show in this case enhanced CDI with T92W Orai1 mutant but not with WT Orai1. Several additional experiments are needed to further strengthen the idea that T92W Orai1 currents inactivate at a lower intracellular Ca^2+^ concentration. If the authors are to use a holding potential of +30mV that is maintained over a significantly longer period of time, to provide sufficient time to decrease intracellular Ca^2+^ concentration, one expects a similar outcome in terms of T92W CDI.

2) The authors should also examine additional buffering conditions utilizing different concentrations of BAPTA during their CDI two-step protocols. For instance, what will the 2-step inactivation protocol look like with 8mM BAPTA or an intermediary concentration between 0.8, 8mM, and 20 mM BAPTA? Since BAPTA is removing also local Ca^2+^, the CDI should probably be not different/less pronounced in the 2nd step. Ideally, CDI should be shown under comparable conditions, i.e. all with identical intracellular Ca^2+^ buffer concentrations. What is the rationale for using 8mM and 0.8mM BAPTA, but 10mM and 20mM EGTA? Why not 10 mM and 20 mM BAPTA?

3) A recent study (PMID: 34796201) reported on the impact of L138F in the Orai channel using MD simulations. They discovered two local but essential conformational changes they supposed to be key to the activation. On the one hand, a rotation of L138 and on the other hand a counter-clockwise rotation of F99, resulting in higher hydration. Could you discuss these findings in addition or even find a correlation with your study? Similarly, is there any evidence in past MD simulations by this group (or others) that L138 and H134 apply to oppose push-pull forces to TM1, as suggested in the present manuscript?

4) The authors discuss the analysis of Figure 6 in terms of a simple reaction scheme, but there are no fitted data or fitted rate constants. Where are the fitted data shown and conformance to the reaction scheme to support the conclusions made by this analysis?

5) Figure 7 – If not from W76, R83, or Y80, please provide insights on where is the additional/residual inactivation coming from after deletion of 267-301.

6) The authors focused on L138F and T92W showing CDI in the absence of STIM1 to conclude that CDI is intrinsic to Orai1. What does CDI look like with additional mutations within the same sites (e.g. L138Y and T92F/Y)? For data in Figure 6, what does the same data look like for the L138F Orai1 mutant?

7) Does enhanced CDI manifest in smaller SOCE in Ca^2+^ measurements? Can the authors show what the Ca^2+^ signal (measured with a dye) looks like for L138F and T92W mutants?

*Reviewer #1 (Recommendations for the authors):*

This is a very clearly written manuscript that reports on novel and interesting findings of relevance to CRAC channel regulation, function, and dysfunction. STIM1 was previously shown to be necessary for CDI of WT Orai1 and the present study sheds light and refines previous models of CDI and hints that the gate for CDI is likely located on the Orai1 channel itself and that STIM1 modulates CDI by altering the Ca^2+^ sensitivity of Orai1 channels. Although the altered Ca^2+^ sensitivity of L138 and T92 mutants and its normalization is novel and enhances our knowledge of the mechanism of CRAC channel CDI, the full picture of CDI and particularly the Ca^2+^ sensing site(s) for CDI within Orai1 remain obscure. This and other comments are listed below:

– The authors focused on L138F and T92W showing CDI in the absence of STIM1 to conclude that CDI is intrinsic to Orai1. This statement needs to be further supported by additional mutations within the same sites (e.g. L138Y and T92F/Y) with similar experiments as in Figure 4 and 5.

– Based on the authors' calculations of intracellular Ca^2+^ under BAPTA vs EGTA, they propose that the reversal of CDI behavior in the different buffer conditions (Figure 5) is due to enhanced Ca^2+^ sensitivity of the mutants and that mutant channels reach equilibrium for CDI at a holding potential of +30 mV, such as that with the classic step protocol CDI is already in place before the hyperpolarizing step. Although this is a sensible and straightforward interpretation and 2-step CDI pulse data in Figure 6 lend it support, this explanation would be more convincing if additional recordings were conducted with mutants with a holding potential of +100mV or +120mV with two different concentrations of BAPTA falling between 0.8 and 8mM and between 8 and 20 mM.

– For data in Figure 6, what does the same data look like for the L138F Orai1 mutant?

– The patch clamp data is of high quality and convincing, but the authors provide no clues as to how the Ca^2+^ signal (measured with a dye, e.g. Fura) looks like for all the L138 and T92 mutants in the presence of different external driving force conditions.

*Reviewer #2 (Recommendations for the authors):*

The manuscript is timely and of broad interest, nevertheless, the authors should address several further points in particular regarding the puzzling behavior of T92W in dependence on Ca^2+^.

The authors state: "L138F currents exhibited rapid rundown during the first 20-30 seconds (Figure 1A)." Is this rundown occurring in a Ca^2+^-dependent manner? Are there other reasons for this rundown? In contrast, T92F does not show inactivation/rundown in the time course (Figure 2E), while T92W does (Figure 3C). What could be the reason for that?

The authors state: "whether relieving this clash by reducing the size of the opposing residue could relieve constitutive channel activity" For this, they incorporated glycine, however, glycine is also known to affect the geometry of the helical transmembrane domain by increasing flexibility. Hence, did the authors observe the "rescue effect" also with an alanine, for instance?

Figure 4: Could you please show CDI always under comparable conditions – all with identical intracellular Ca^2+^ buffer concentrations?

What is the rationale for using 8mM and 0.8mM BAPTA, but 10mM and 20mM EGTA? Why not 10 mM and 20 mM BAPTA?

The authors propose that the Ca^2+^ sensitivity of the inactivation of T92W is substantially increased. Constitutive Orai1 T92W currents inactivate at lower Ca^2+^ concentrations compared to Orai1 WT. The authors applied a sophisticated two-step protocol using a prepulse to +100 mV to reveal subsequently increased inactivation of T92W Orai1 in 10 mM EGTA which is not seen with Orai1 WT. I would expect a similar effect if the prepulse goes to +30mV and is applied for a longer time interval providing sufficient time to decrease intracellular Ca^2+^ concentrations. Additionally, what will the 2-step inactivation protocol look like for 8mM BAPTA? Since BAPTA is removing also local Ca^2+^, the CDI should probably be not different/less pronounced in the 2nd step.

A recent study (PMID: 34796201) reported on the impact of L138F in the Orai channel using MD simulations. They discovered two local but essential conformational changes they supposed to be key to the activation. On the one hand, a rotation of L138 and on the other hand a counter-clockwise rotation of F99, resulting in higher hydration. Could you discuss these findings in addition or even find a correlation with your study?

The idea that the Ca^2+^ sensor is likely located within the Orai1 protein itself is compatible with the T92W/L138F mutants, but how will it explain the missing CDI of Orai1 P245L?

*Reviewer #3 (Recommendations for the authors):*

Tryptophan is one of three aromatic amino acids that contain a benzene ring in their side chains; the other two being Phe and Tyr. Thus, the claim that "the introduction of a large benzene ring at L138 likely leads to a steric clash of the exogenous Phe or Tyr side-chains with residues in TM1 causing channel activation" seems inconsistent with their Trp substitution. In the case of Trp, the benzene is fused to a pyrrole ring.

In all hydrophobicity plots, His could have two points (protonated and deprotonated); the authors should specify which one is plotted and why they believe this is the protonation state in their electrophys measurements.

Supplementary Figure 3 – Not only did R and D result in a loss of function but also G and P. The authors should comment on why they think this may be the case.

Figure 4 – Why isn't the same intracellular Ca^2+^ chelator used for WT and mutants in this set of data?

In Figure 4B and D, are the DVF panels mixed up, or are the DVF plots in 4C and 4E mixed up? – In the current layout, it appears that L138F shows a time-dependent inactivation in the DVF buffer whereas the T92W does not, but what is the basis for this inactivation if buffers are DVF? Statistical comparisons need to be made in 4C and 4E and differences interpreted.

The authors should describe/discuss the data in Figure 5E.

It is not totally clear to me why comparisons between WT and T92W mutant are not being made at 8 mM BAPTA chelator for both. The equations are comparing 10 mM EGTA (WT) with 8 mM BAPTA (T92W). Does the increased Ca^2+^ sensitivity of T92W hold when doing these comparisons using the same chelator?

Page 18 – The authors discuss the analysis of Figure 6 in terms of a simple reaction scheme, but I don't see any fitted data or any fitted rate constants. Where is the fitted data shown and conformance to the reaction scheme to support the conclusions made by this analysis?

Figure 7 – If not from W76, R83, and Y80 where is the additional/residual inactivation coming from after deletion of 267-301? What happens if you delete the N- and C-termini; do you completely abrogate CDI?

Figure 8 – Is it possible that Ca^2+^ interactions with STIM1, specifically the ID domain, may be buffering and normalizing the CDI response? Can this be tested with ID domain mutants?

Figure 8 – Supplement 1 – Can the authors comment on why the voltage dependence of CDI is lost for the Y80E/T92W mutant, even in the presence of STIM1?

Figure 9 – Is there any evidence in past MD simulations by this group that L138 and H134 apply to oppose push-pull forces to TM1?

Some sample double-exponential fits of the inactivation profiles should be shown for the mutants and WT Orai1 channels. Differences in the goodness of fits may point to differences in CDI mechanisms.

It is suggested that "a change in Ca^2+^ sensitivity of CDI could presumably occur via STIM1-driven change in the conformation of the domain containing the Ca^2+^ binding site at the Orai1 C-terminus." The authors should consider and discuss how Ca^2+^ binding to the ID when STIM1 is coupled to Orai1 channels may affect the Ca^2+^ sensitivity of CDI.

---

## [Author Response]

Essential revisions:1) Additional experiments are required to strengthen data related to CDI of the T92W Orai1 mutant. The authors propose that constitutive T92W Orai1 currents inactivate at a lower intracellular ca^2+^ concentration compared to WT Orai1. Based on a 2-step CDI protocol from a holding membrane potential of +100mV and 10mM EGTA in the patch pipette, the authors show in this case enhanced CDI with T92W Orai1 mutant but not with WT Orai1. Several additional experiments are needed to further strengthen the idea that T92W Orai1 currents inactivate at a lower intracellular ca^2+^ concentration. If the authors are to use a holding potential of +30mV that is maintained over a significantly longer period of time, to provide sufficient time to decrease intracellular ca^2+^ concentration, one expects a similar outcome in terms of T92W CDI.

We thank the reviewer for this comment. Increasing the duration of recovery at +30 mV does not make any difference (Author response image 1) as recovery from CDI occurs much faster than the 200 ms interpulse time interval that we used in Figure 6 of the paper. For native CRAC channels, Zweifach and Lewis (1995) previously showed that recovery from CDI occurs over a biexponential time course with time constants of 7 and 75 ms, and is nearly complete by ~200 ms (in their recording, the recovery potential was -12 mV compared to +30 mV in our experiments). For T92W Orai1, we have similarly found that recovery from CDI also occurs rapidly and reaches steady-state by 120 ms at +100 mV (Author response image 1).

**Author response image 1. sa2fig1:** Recovery from CDI of T92W Orai1 is not improved by prolonging the interpulse interval. (A) The traces show inactivation in response to -100 mV pulses separated by a 3 s depolarizing step to +30 mV to promote recovery from CDI. Extracellular ca^2+^ was 20 mM and the intracellular solution contained 10 mM EGTA. (B) Recovery time course of CDI of T92W Orai1 channels. The interpulse interval was varied from 40 ms to 200 ms. The recovery time course is shown on the right plot. Recovery reaches steady-state by 200 ms.

We explicitly tested the reviewer’s suggested experiment by extending the recovery duration at +30 mV to 3 s (> 10-fold longer than the original 200 ms interpulse interval) (Author response image 1) but this offered no added recovery. This result is consistent with the notion that channels are already at equilibrium between open and inactivated states within 200 ms (with the equilibrium set by the Ca influx occurring at +30 mV). To promote recovery, what is needed is to reduce submembrane [Ca] *further* at the recovery potential, which can only be achieved by further depolarizing the recovery potential to reduce the driving force for Ca^2+^ influx and therefore the submembrane [Ca^2+^] below what occurs at +30 mV. This is what we tried to accomplish by using the +100 mV recovery pulse.

Also, just to clarify, the above statement in the comment “the authors show enhanced CDI with T92W Orai1 mutant but not with WT Orai1” is actually not correct. What we show is enhanced *recovery* from CDI in T92W (not enhanced CDI) compared to WT Orai1 (because T92W Orai1 channels enter the inactivation state *faster* at the recovery potential of +100 mV than WT).

2) The authors should also examine additional buffering conditions utilizing different concentrations of BAPTA during their CDI two-step protocols. For instance, what will the 2-step inactivation protocol look like with 8mM BAPTA or an intermediary concentration between 0.8, 8mM, and 20 mM BAPTA? Since BAPTA is removing also local Ca^2+^, the CDI should probably be not different/less pronounced in the 2nd step. Ideally, CDI should be shown under comparable conditions, i.e. all with identical intracellular Ca^2+^ buffer concentrations. What is the rationale for using 8mM and 0.8mM BAPTA, but 10mM and 20mM EGTA? Why not 10 mM and 20 mM BAPTA?

Done. We added the results for 8 mM BAPTA data to Figure 6 —figure supplement 1. The new data show that there is still recovery that happens in BAPTA but it is less than that seen in EGTA as the membrane holding potential is further depolarized. This is because with BAPTA, there is still substantial CDI that occurs during the hyperpolarizing steps indicating that at this level of buffering, resting inactivation at the holding potential (+30 mV) is less than what occurs in EGTA. As a result of *reduced* resting inactivation, there is therefore less need for the membrane potential to be further depolarized to promote recovery from CDI.

Rationale for the BAPTA and EGTA concentrations. We apologize for the confusion regarding the use of the different buffer concentrations. We recognize that the order of the buffer concentrations shown in the original Figure 5 of the paper was confusing. To address this issue, we have now added data for 20 mM BAPTA to Figure 5A and also show CDI at exactly the same buffer concentration for both WT Orai1 and T92W Orai1. The concentration of the buffers are also ordered from low to high for each buffer (Figure 5A of the manuscript).

We should note that the concentrations of EGTA and BAPTA were not chosen at random but to contrast the effective buffering properties of 10 mM EGTA and 8 mM BAPTA. The concentrations of the buffers were chosen to match the local buffering created by the buffers. As Neher (Neher, 1986; Neher and Augustine, 1992) showed, the effective local buffering is mainly dictated by the space constant of Ca^2+^ diffusion under the different buffering conditions. The space constant is given by the relation: λ=(DCakonB)1/2 where *D_Ca_* is the Ca^2+^ diffusion coefficient (~3 x 10-10 m^2^ s^-1^), k_on_ is the forward rate constant of the buffer (*k_on_* = 6 x 10^8^M^-1^s^-1^ for BAPTA and 1.5 x 10^6^ M^-1^s^-1^ for EGTA) at pH 7.2. *B* is the concentration of the buffer. *λ* is 7.9 nm in the presence of 8 mM BAPTA and much larger at 25 nm at 0.8 mM BAPTA. At 20 mM BAPTA, *λ* drops to 5 nm. Thus, changing the BAPTA concentration from 8 mM to 0.8 mM markedly increases the capture distance for a Ca^2+^ ion, effectively putting it beyond the molecular dimensions of a typical ion channel (for context, the Orai1 channel diameter is ~6-7 nm (Hou et al., 2012)). Once *λ* is beyond the molecular dimensions of the channel, we speculate that the local Ca^2+^ will be essentially unbuffered for CDI. In EGTA solutions, *λ* is 141 nm at 10 mM EGTA, and 100 nm at 20 mM EGTA. These distances are far too large for EGTA to have *any* meaningful local buffering capacity. Thus, CDI is essentially unaffected by EGTA in the 10-20 mM range (and hence not very meaningful to use these EGTA concentrations for buffering local Ca^2+^ that drives CDI).

**Author response image 2. sa2fig2:** Intracellular [Ca^2+^] profiles from a point source of Ca^2+^ entry. The left plots show [Ca^2+^] profiles in varying concentrations of EGTA and the right profiles in BAPTA. Whereas local Ca^2+^ is largely unaffected by variations in [EGTA], they are profoundly reduced by increasing [BAPTA] from 0.8 mM to 8 mM or more. The dotted line shows the estimated distance of the putative Ca^2+^ binding site for CDI (Zweifach and Lewis, 1995. JGP).

We note that for T92W Orai1, its enhanced Ca^2+^ sensitivity to CDI means that CDI is essentially maximal and reaches equilibrium with EGTA at the holding potential itself, such as membrane hyperpolarization elicits no further CDI. With BAPTA, buffering becomes increasingly effective as the concentration is raised such that CDI is visible and apparent during hyperpolarizing steps. Hence, we empirically selected buffer concentrations to span the widest possible range of space constants.

3) A recent study (PMID: 34796201) reported on the impact of L138F in the Orai channel using MD simulations. They discovered two local but essential conformational changes they supposed to be key to the activation. On the one hand, a rotation of L138 and on the other hand a counter-clockwise rotation of F99, resulting in higher hydration. Could you discuss these findings in addition or even find a correlation with your study? Similarly, is there any evidence in past MD simulations by this group (or others) that L138 and H134 apply to oppose push-pull forces to TM1, as suggested in the present manuscript?

Thank you for raising this point. Previous MD simulations performed by Zhang et al. suggest that in dOrai L210F (L138F Orai1), the introduction of the Phe side-chain at position 210 stabilizes the side chain in a clockwise rotated state compared to in WT channels. This conformational change is associated with opening of the hydrophobic gate in the pore through counter-clockwise rotation of TM1 as previously shown (Yamashita et al., 2017; Yeung et al., 2018; Bulla et al., 2019) as well as inner pore dilation as shown in other constitutively active channels (Hou et al., 2020; Liu et al., 2019; Dong et al., 2019). Although our current study does not directly examine whether or not the L138F side-chain is rotated compared to the WT L138 residue, both studies implicate that L138-TM1 interactions are critical for determining the outer and inner pore conformations. Based on mutational analysis, we hypothesize that this effect is governed by intersubunit L138-T92 steric interactions (Figure 9C). In this context, a clockwise rotation of L138F as suggested by Zhang et al. would increase the steric interaction and is fully compatible with our data. We have now revised the text to include this information (p 28-29 of Discussion).

4) The authors discuss the analysis of Figure 6 in terms of a simple reaction scheme, but there are no fitted data or fitted rate constants. Where are the fitted data shown and conformance to the reaction scheme to support the conclusions made by this analysis?

We appreciate this concern, but actually this is not currently possible since we do not have any realistic measures of the rate constants for entry of channels into inactivated states or the recovery of channels from inactivated states (into open or other closed states). The simple scheme with one closed state, one open state, and one inactivated state that we used in the manuscript was done purely for conceptual understanding of what is *likely* to happen when the occupancy of channels in different states is altered. The real-life situation is likely to be far more complex with multiple ca^2+^ binding steps (for each of the six subunits), multiple STIM1 binding steps to the channel, and potentially several inactivated states (as already suggested by the presence of more than one exponential for the CDI process both in WT and in T92W channels). We used the scheme only to conceptually illustrate the fundamental steps of CDI and what might happen if channel occupancy in the inactivated state increases. It is not meant to be a quantitative reproduction of the data.

However, we do realize that this creates confusion in the paper, and in response to this concern, we have moved the state diagram scheme and its interpretation to the Discussion section (from the Results).

5) Figure 7 – If not from W76, R83, or Y80, please provide insights on where is the additional/residual inactivation coming from after deletion of 267-301.

There are some acidic residues in the loop which could be potential sites. Moreover, our previous work has also suggested a role for Ca binding within the pore itself (Yamashita et al., 2007). Those are possibilities. The role of these sites for T92W CDI will be addressed in followup studies. We would like to note that while we do understand and appreciate the desire to elucidate the basis of the residual inactivation that is left when the c-terminus is truncated, these studies are outside the scope of this manuscript. Our study already contains a huge amount of experimental analysis with over 200 mutations analyzed by traditional patch-clamp analysis. We began with molecular dissection of a pathological human mutation, dissected the basis of its constitutive activation phenotype, then analyzed the unusual CDI in these mutants. As our paper already contains a large amount of experimental work and novel conceptual findings, we hope reviewers will understand that elucidating the basis of the inactivation or the precise molecular role of STIM1 in CDI is beyond the scope of this manuscript and will be followed up in subsequent work.

6) The authors focused on L138F and T92W showing CDI in the absence of STIM1 to conclude that CDI is intrinsic to Orai1. What does CDI look like with additional mutations within the same sites (e.g. L138Y and T92F/Y)? For data in Figure 6, what does the same data look like for the L138F Orai1 mutant?

We have added data for L138Y and T92F/Y in Figure 4 Figure Supplement 1. Data for L138F is also now added in Figure 6 —figure supplement 1.

7) Does enhanced CDI manifest in smaller SOCE in ca^2+^ measurements? Can the authors show what the Ca^2+^ signal (measured with a dye) looks like for L138F and T92W mutants?

No, T92W does not produce lower [Ca^2+^] because the mutation elicits such a large constitutively active channel that basal [Ca^2+^]i is very high in T92W expressing HEK cells. [Ca^2+^] levels are also regulated by the synergistic action of Ca influx pathways, pumps, mitochondria, and other processes that collectively are going to be engaged to different extents in WT and T92W expressing cells. The traces in Author response image 3 show the fura-2 Ca imaging done in untransfected, WT, L138F and T92W expressing cells. As can be seen the very high constitutive activity of T92W elevates cytosolic Ca to high levels which is not necessarily reduced by CDI. We do not know the physiological implications of the increased inactivation. Our study rather deals with the molecular mechanism of the process which remains unknown.

**Author response image 3. sa2fig3:** Intracellular [Ca^2+^] in HEK293 cells expressing T92W Orai1, L138F Orai1, WT Orai1 + STIM1, or WT Orai1 alone. Intracellular [Ca^2+^] was measured using fura-2. Stores were depleted as indicated using thapsigargin and extracellular Ca^2+^ added back to assess SOCE.

Reviewer #1 (Recommendations for the authors):This is a very clearly written manuscript that reports on novel and interesting findings of relevance to CRAC channel regulation, function, and dysfunction. STIM1 was previously shown to be necessary for CDI of WT Orai1 and the present study sheds light and refines previous models of CDI and hints that the gate for CDI is likely located on the Orai1 channel itself and that STIM1 modulates CDI by altering the Ca^2+^ sensitivity of Orai1 channels. Although the altered Ca^2+^ sensitivity of L138 and T92 mutants and its normalization is novel and enhances our knowledge of the mechanism of CRAC channel CDI, the full picture of CDI and particularly the Ca^2+^ sensing site(s) for CDI within Orai1 remain obscure. This and other comments are listed below:– The authors focused on L138F and T92W showing CDI in the absence of STIM1 to conclude that CDI is intrinsic to Orai1. This statement needs to be further supported by additional mutations within the same sites (e.g. L138Y and T92F/Y) with similar experiments as in Figure 4 and 5.– Based on the authors' calculations of intracellular Ca^2+^ under BAPTA vs EGTA, they propose that the reversal of CDI behavior in the different buffer conditions (Figure 5) is due to enhanced Ca^2+^ sensitivity of the mutants and that mutant channels reach equilibrium for CDI at a holding potential of +30 mV, such as that with the classic step protocol CDI is already in place before the hyperpolarizing step. Although this is a sensible and straightforward interpretation and 2-step CDI pulse data in Figure 6 lend it support, this explanation would be more convincing if additional recordings were conducted with mutants with a holding potential of +100mV or +120mV with two different concentrations of BAPTA falling between 0.8 and 8mM and between 8 and 20 mM.– For data in Figure 6, what does the same data look like for the L138F Orai1 mutant?

Thank you for the suggestion. As recommended, we have carried out recordings of T92W in the presence of 8 mM BAPTA (again this concentration was chosen to be consistent with the rest of the data). These new results are shown Figure 6 —figure supplement 1. The data shows that recovery of T92W in the presence of BAPTA looks much more like WT Orai1 (in the presence of EGTA) than T92W in EGTA. This is very much consistent with the finding that T92W Orai1 shows enhanced Ca^2+^ sensitivity to CDI and inactivation is primarily only seen in the presence of BAPTA.

We also carried out new recordings of paired pulse recovery for the L138F mutant and these data are shown in Figure 6 —figure supplement 1C,D. The recovery of L138F also looks much more like WT Orai1 rather than T92W Orai1. We think this is because the current density is so small, that the degree of resting inactivation is very little. Hence the holding potential of +30 mV is sufficient to promote recovery from CDI.

– The patch clamp data is of high quality and convincing, but the authors provide no clues as to how the Ca^2+^ signal (measured with a dye, e.g. Fura) looks like for all the L138 and T92 mutants in the presence of different external driving force conditions.

T92W does not produce lower [Ca^2+^] because the mutation elicits such a large constitutively active channel that basal [Ca^2+^]i is very high in T92W expressing HEK cells. [Ca^2+^] levels are also regulated by the synergistic action of Ca influx pathways, pumps, mitochondria, and other processes that collectively are going to be engaged to different extents in WT and T92W expressing cells. Figure 4 of the Essential Revisions section above shows fura-2 Ca imaging done in untransfected HEK293 cells and WT Orai1, L138F Orai1 and T92W Orai1 expressing cells. As can be readily seen, the very high constitutive activity of T92W elevates cytosolic Ca to high levels which is not necessarily reduced by CDI.

Reviewer #2 (Recommendations for the authors):The manuscript is timely and of broad interest, nevertheless, the authors should address several further points in particular regarding the puzzling behavior of T92W in dependence on Ca^2+^.The authors state: "L138F currents exhibited rapid rundown during the first 20-30 seconds (Figure 1A)." Is this rundown occurring in a Ca^2+^-dependent manner? Are there other reasons for this rundown? In contrast, T92F does not show inactivation/rundown in the time course (Figure 2E), while T92W does (Figure 3C). What could be the reason for that?

As shown in Figure 4 Figure Supplement 1, CDI of T92W Orai1 is much more prominent than the CDI of T92F (which only shows minimal CDI). Hence the rundown basically reflects accumulation of inactivation and lack of recovery at the holding potential which is more prominent in T92W than T92F.

The authors state: "whether relieving this clash by reducing the size of the opposing residue could relieve constitutive channel activity" For this, they incorporated glycine, however, glycine is also known to affect the geometry of the helical transmembrane domain by increasing flexibility. Hence, did the authors observe the "rescue effect" also with an alanine, for instance?

We thank the reviewer for noting this. The Gly substitution at T92 does not impair channel gating. In response to the reviewer concern, we tested the ability of L138F/T92G Orai1 to be gated by STIM1. Although not constitutively open, in the presence of STIM1, L138F/T92G is still activated and gated by STIM1 to yield large CRAC currents (new data, Figure 2 —figure supplement 1A).

This result indicates that the Gly mutant is indeed functional and the mutation does not significantly affect Orai1 structure to affect function.

As suggested, we also generated a T92A/L138F double mutant and analyzed it phenotype. However, this double mutant is constitutively active. This result is not surprising since the Ala side-chain has a much larger volume and mass compared to Gly and therefore cannot reverse the GOF phenotype of the L138F mutant.

**Author response image 4. sa2fig4:** Introduction of an Ala residue at T92 causes constitutive activation of T92A/L138F Orai1. T92A/L138F Orai1 currents were measured in the absence of STIM1 using standard methods. The left plot shows the peak current at -100 mV plotted over time and the right graph shows the current-voltage relationship of the ca^2+^ and monovalent currents as indicated.

Figure 4: Could you please show CDI always under comparable conditions – all with identical intracellular Ca^2+^ buffer concentrations? What is the rationale for using 8mM and 0.8mM BAPTA, but 10mM and 20mM EGTA? Why not 10 mM and 20 mM BAPTA?

We apologize for not making this clearer but these concentrations were chosen empirically to achieve the widest variation in the local buffering capacity for Ca^2+^. As explained in point 2 of the essential revisions’ response, the concentrations of the buffers were chosen to span a range of local buffering capacities. As Neher (Neher, 1986; Neher and Augustine, 1992) showed, the effective local buffering is mainly dictated by the space constant of Ca^2+^ diffusion under the different buffering conditions. The space constant is given by the relation: λ=(DCakonB)1/2 where *D_Ca_* is the Ca^2+^ diffusion coefficient (~3 x 10-10 m^2^ s^-1^), k_on_ is the forward rate constant of the buffer (*k_on_* = 6 x 10^8^M^-1^s^-1^ for BAPTA and 1.5 x 10^6^ M^-1^s^-1^ for EGTA) at pH 7.2. *B* is the concentration of the buffer. *λ* is 7.9 nm in the presence of 8 mM BAPTA and much larger at 25 nm at 0.8 mM BAPTA. At 20 mM BAPTA, *λ* drops to 5 nm. Thus, changing the BAPTA concentration from 8 mM to 0.8 mM markedly increases the capture distance for a Ca^2+^ ion, effectively putting it beyond the molecular dimensions of a typical ion channel (for context, the Orai1 channel diameter is ~6-7 nm (Hou et al., 2012)). Once *λ* is beyond the molecular dimensions of the channel, we speculate that the local Ca^2+^ will be essentially unbuffered for CDI. In EGTA solutions, *λ* is 141 nm at 10 mM EGTA, and 100 nm at 20 mM EGTA. These distances are far too large for EGTA to have *any* meaningful local buffering capacity. Thus, CDI is essentially unaffected by EGTA in the 1020 mM range (and hence not very meaningful to use these concentrations for EGTA).

We do recognize that the order of the buffer concentrations as shown in the original Figure 5 of the paper was confusing. To address this issue, we now show CDI at exactly the same buffer concentration for both WT Orai1 and T92W Orai1. We have also added data for 20 mM BAPTA and changed the figure so that the buffers are ordered from low-to-high for each buffer (Figure 5).

In the case of T92W, its enhanced Ca^2+^ sensitivity means that CDI is essentially maximal in EGTA internal solutions and reaches equilibrium at the holding potential itself, such as membrane hyperpolarization can elicit no further CDI. With BAPTA, buffering becomes increasingly effective as the buffer concentration is increased, such that CDI is visible and apparent during hyperpolarizing steps. Hence, we selected buffer concentrations to span the widest possible range of space constants.

The authors propose that the Ca^2+^ sensitivity of the inactivation of T92W is substantially increased. Constitutive Orai1 T92W currents inactivate at lower Ca^2+^ concentrations compared to Orai1 WT. The authors applied a sophisticated two-step protocol using a prepulse to +100 mV to reveal subsequently increased inactivation of T92W Orai1 in 10 mM EGTA which is not seen with Orai1 WT. I would expect a similar effect if the prepulse goes to +30mVm and is applied for a longer time interval providing sufficient time to decrease intracellular Ca^2+^ concentrations. Additionally, what will the 2-step inactivation protocol look like for 8mM BAPTA? Since BAPTA is removing also local Ca^2+^, the CDI should probably be not different/less pronounced in the 2nd step.

Increasing the duration at +30 mV does not make any difference as recovery from inactivation occurs over time scales that are much faster than the 200 ms inter-pulse time as shown in Figure 1 in the essential revisions response. This is consistent with the very early findings of Zweifach and Lewis (1995) who showed that recovery from CDI occurs with time constants of 7 and 75 ms and is nearly complete by ~200 ms (in their recording, the recovery potential was -12 mV compared to +30 mV or +100 mV in our experiments) (Zweifach and Lewis, 1995). In our tests, we have found that recovery of CDI in T92W also occurs rapidly and equilibrates with inactivation. Extending the duration at +30 mV to 3 s (see below) offered no added benefit as the channels are already at equilibrium between open and inactivated states (with the equilibrium set by the Ca influx occurring at +30 mV). What was needed is to reduce submembrane [Ca] further, which could only be achieved by furthering depolarizing the recovery potential to reduce the driving force for Ca^2+^ influx and therefore the submembrane [Ca^2+^] even below what occurs at +30 mV. This is what we tried to accomplish by using the +100 mV recovery pulse.

We have added the results for 8 mM BAPTA data into Figure 6 —figure supplement 1A. The new data show that there is still recovery that happens in BAPTA but it is less than that seen in EGTA as the membrane holding potential is further depolarized. This is because with BAPTA, there is still substantial CDI that occurs during the hyperpolarizing steps indicating that with this buffering, resting inactivation at the holding potential (+30 mV) is less than what occurs in EGTA. As a result of reduced resting inactivation, there is therefore less need for the membrane potential to be further depolarized to promote recovery from CDI.

A recent study (PMID: 34796201) reported on the impact of L138F in the Orai channel using MD simulations. They discovered two local but essential conformational changes they supposed to be key to the activation. On the one hand, a rotation of L138 and on the other hand a counter-clockwise rotation of F99, resulting in higher hydration. Could you discuss these findings in addition or even find a correlation with your study?

In this MD simulation study by Zhang et al., the authors found that in the dOrai L210F (human L138F Orai1) mutant, the introduced Phe side-chain is stabilized in a clockwise rotated state compared to in WT channels. This conformational change is associated with opening of the hydrophobic gate in the pore through counter-clockwise rotation of TM1 as previously proposed (Bulla et al., 2019; Yamashita et al., 2017; Yeung et al., 2018) as well as inner pore dilation as shown in other constitutively active channels (Dong et al., 2019; Frischauf et al., 2017; Hou et al., 2020). Although our current study does not directly examine whether or not the L138F side-chain is rotated compared to the WT L138 residue, both studies implicate L138-TM1 interactions are critical for determining the outer and inner pore conformations. Based on mutational analysis, we hypothesize that this effect is governed by intersubunit L138-T92 steric interactions (Figure 9C). In this context, a clockwise rotation of L138F as suggested by Zhang et al. would increase the steric interaction and is fully compatible with our data. We have now revised the Discussion to include this information.

The idea that the Ca^2+^ sensor is likely located within the Orai1 protein itself is compatible with the T92W/L138F mutants, but how will it explain the missing CDI of Orai1 P245L?

We agree that this is a conundrum that needs more work to be understood. Orai1 P245L channels, like all other previously reported gain-of-function mutants (e.g. H134), do not exhibit CDI but rather have stable or slightly potentiating current over 100 ms hyperpolarization steps. This demonstrates that Orai1 can be activated through mutation of other TM2-4 residues without exhibiting inactivation. To our knowledge, the T92-L138 locus is the only area within Orai1 reported to date where mutations can both open the channel and unmask STIM-independent inactivation. It is not clear whether the unique ability of T92-L138 mutants to inactivate is due to trapping the channel in a conformation further “downstream” to the STIM1-gated state or whether activation and inactivation are controlled by two separate pathways. However, we hypothesize that the proximity of the T92-L138 locus to inner pore residues which play a key role in regulating CDI (Mullins et al., 2016, Figure 7), contributes to this process.

Reviewer #3 (Recommendations for the authors):Tryptophan is one of three aromatic amino acids that contain a benzene ring in their side chains; the other two being Phe and Tyr. Thus, the claim that "the introduction of a large benzene ring at L138 likely leads to a steric clash of the exogenous Phe or Tyr side-chains with residues in TM1 causing channel activation" seems inconsistent with their Trp substitution. In the case of Trp, the benzene is fused to a pyrrole ring.

Thank you for raising this point. We also initially found the result of Trp to be puzzling. However, we later found that local protein conformation surrounding T92 appears to be very sensitive to the exact locations of the side chain atoms. For example, T92L and T92I exhibit different levels of inactivation despite being the same size. We have also performed other T92-L138 double mutant experiments and seen that the phenotypes are sensitive to the positions of individual atoms (e.g. T92C/L138S is different from T92S/L138C). Because the benzene ring of Trp is fused distally to the pyrrole ring, this might create enough distance away from the α atom to avoid the steric clash seen in Phe and Tyr.

In all hydrophobicity plots, His could have two points (protonated and deprotonated); the authors should specify which one is plotted and why they believe this is the protonation state in their electrophys measurements.

Because we did not actively try to control the protonation state of introduced histidines, we cannot be certain the fraction of protonated vs. deprotonated forms. However, since T92-L138 is closer to the intracellular surface (pH ~7), it is more likely in deprotonated form and possibly forming hydrogen bonds with neighboring residues.

Supplementary Figure 3 – Not only did R and D result in a loss of function but also G and P. The authors should comment on why they think this may be the case.

In the case of Gly and Pro mutations, one possibility is that these substitutions disrupt the α helical structure of the transmembrane domains.

Figure 4 – Why isn't the same intracellular Ca^2+^ chelator used for WT and mutants in this set of data?

The data for similar concentrations of EGTA and BAPTA are actually shown in Figure 5. In Figure 4, we used conditions that showed CDI in the WT and T92W mutants as this is the very first introduction to the idea that the T92W mutant shows CDI. In Figure 5, we present the results at the same concentrations of EGTA and BAPTA for both mutants. In EGTA, T92W CDI is essentially maxed out and channels are essentially at equilibrium between inactivated and open states, whereas in BAPTA which is a stronger/better buffer, inactivation is restored and clearly visible during hyperpolarizing steps. This is explained in terms of the enhanced Ca-sensitivity for CDI of T92W Orai1 channels.

In Figure 4B and D, are the DVF panels mixed up, or are the DVF plots in 4C and 4E mixed up? – In the current layout, it appears that L138F shows a time-dependent inactivation in the DVF buffer whereas the T92W does not, but what is the basis for this inactivation if buffers are DVF? Statistical comparisons need to be made in 4C and 4E and differences interpreted.

No the plots are as shown and not mixed up. There is no inactivation at all in DVF solutions in any of the conditions, consistent with the Ca^2+^ dependence of the CDI process. The errors bars in Figure 4 are indeed present and if not visible, they are smaller than the size of the dots. We have added statistics to the data as suggested.

The authors should describe/discuss the data in Figure 5E.It is not totally clear to me why comparisons between WT and T92W mutant are not being made at 8 mM BAPTA chelator for both. The equations are comparing 10 mM EGTA (WT) with 8 mM BAPTA (T92W). Does the increased Ca^2+^ sensitivity of T92W hold when doing these comparisons using the same chelator?

We apologize for not making this clearer. This was done to provide the widest variation in the local buffering capacity of the buffer used. Please see our response to point 2 of essential revisions and to reviewer 2 for detailed explanation of this.

Page 18 – The authors discuss the analysis of Figure 6 in terms of a simple reaction scheme, but I don't see any fitted data or any fitted rate constants. Where is the fitted data shown and conformance to the reaction scheme to support the conclusions made by this analysis?

As explained in the essential revisions, this is not actually currently possible with the available information since we do not have any realistic measures of the rate constants for entry of channels into inactivation states or the recovery of channels from inactivation states (into open or other closed states). The simple scheme with one closed, one open, and one inactivated state that we used in the manuscript was done purely for conceptual understanding of what is likely to happen when the occupancy of channels in different states is altered. The real-life situation is likely to be far more complex with multiple Ca^2+^ binding steps (for each of the six subunits), multiple STIM binding steps to the channel, and potentially several inactivated states (as already suggested by the presence of more than one exponential for the CDI process both in WT and in T92W channels). The Scheme was provided to illustrate what is likely to be the fundamental steps of CDI and what might happen if the occupancy in the inactivated states increases. It is not meant to be a quantitative reproduction of the data.

We have moved the scheme to the Discussion (from the Results) to avoid confusion on this issue.

Figure 7 – If not from W76, R83, and Y80 where is the additional/residual inactivation coming from after deletion of 267-301? What happens if you delete the N- and C-termini; do you completely abrogate CDI?

There are acidic residues in the cytosolic loop connecting TM2 to TM3 which could be potential sites. Moreover, our previous work has also suggested a role for Ca binding within the pore itself (Yamashita et al., 2007). Those are possibilities. The role of these sites for T92W CDI will be addressed in follow-up studies.

We have deleted the N-terminus and as seen with the K85E mutation, deleting the N-terminus abolishes T92W channel gating and there is no current left to study. This is shown in Figure 7 —figure supplement 1A,C.

Figure 8 – Is it possible that Ca^2+^ interactions with STIM1, specifically the ID domain, may be buffering and normalizing the CDI response? Can this be tested with ID domain mutants?

This is certainly a possibility. Tests of the ID mutants will be the topic of a future study and we definitely plan to mutate out the ID region to examine STIMs ability to normalize CDI. We want to share that our preliminary tests indicate that the regulation by STIM1 is not as simple as the ID domain and there appear to be contributions from outside the ID region. These studies are outside the scope of the paper and will be systematically addressed in the next study.

Figure 8 – Supplement 1 – Can the authors comment on why the voltage dependence of CDI is lost for the Y80E/T92W mutant, even in the presence of STIM1?

There is currently no MD simulation evidence, and these need to be done, but please also see additional response to Essential revisions point 3.

Figure 9 – Is there any evidence in past MD simulations by this group that L138 and H134 apply to oppose push-pull forces to TM1?Some sample double-exponential fits of the inactivation profiles should be shown for the mutants and WT Orai1 channels. Differences in the goodness of fits may point to differences in CDI mechanisms.It is suggested that "a change in Ca^2+^ sensitivity of CDI could presumably occur via STIM1-driven change in the conformation of the domain containing the Ca^2+^ binding site at the Orai1 C-terminus." The authors should consider and discuss how Ca^2+^ binding to the ID when STIM1 is coupled to Orai1 channels may affect the Ca^2+^ sensitivity of CDI.

Thank you for the suggestion, we have added a Supplementary Figure (Figure 4, figure Supplement 2) with the fits. We have also incorporated some additional discussion of Ca^2+^ binding to the STIM1 ID domain into the Discussion but as noted, this is the topic of an ongoing study, and this question will be addressed in detail in a follow-up paper.

References

Bulla, M., Gyimesi, G., Kim, J.H., Bhardwaj, R., Hediger, M.A., Frieden, M., and Demaurex, N. (2019). ORAI1 channel gating and selectivity is differentially altered by natural mutations in the first or third transmembrane domain. J Physiol 597, 561-582.

Dong, H., Zhang, Y., Song, R., Xu, J., Yuan, Y., Liu, J., Li, J., Zheng, S., Liu, T., Lu, B., Wang, Y., and Klein, M.L. (2019). Toward a Model for Activation of Orai Channel. iScience 16, 356-367.

Frischauf, I., Litvinukova, M., Schober, R., Zayats, V., Svobodova, B., Bonhenry, D., Lunz, V., Cappello, S., Tociu, L., Reha, D., Stallinger, A., Hochreiter, A., Pammer, T., Butorac, C., Muik, M., Groschner, K., Bogeski, I., Ettrich, R.H., Romanin, C., and Schindl, R. (2017). Transmembrane helix connectivity in Orai1 controls two gates for calcium-dependent transcription. Sci Signal 10.

Hou, X., Outhwaite, I.R., Pedi, L., and Long, S.B. (2020). Cryo-EM structure of the calcium release-activated calcium channel Orai in an open conformation. *ELife* 9.

Hou, X., Pedi, L., Diver, M.M., and Long, S.B. (2012). Crystal Structure of the Calcium ReleaseActivated Calcium Channel Orai. Science 338, 1308-1313.

Neher, E. (1986). Concentration profiles of intracellular calcium in the presence of a diffusible chelator.. Experimental Brain Research Series 14, 80-96.

Neher, E., and Augustine, G.J. (1992). Calcium gradients and buffers in bovine chromaffin cells. J Physiol 450, 273-301.

Yamashita, M., Navarro-Borelly, L., McNally, B.A., and Prakriya, M. (2007). Orai1 mutations alter ion permeation and ca^2+^-dependent fast inactivation of CRAC channels: evidence for coupling of permeation and gating. J Gen Physiol 130, 525-540.

Yamashita, M., Yeung, P.S., Ing, C.E., McNally, B.A., Pomes, R., and Prakriya, M. (2017). STIM1 activates CRAC channels through rotation of the pore helix to open a hydrophobic gate. Nat Commun 8, 14512.

Yeung, P.S., Yamashita, M., Ing, C.E., Pomes, R., Freymann, D.M., and Prakriya, M. (2018). Mapping the functional anatomy of Orai1 transmembrane domains for CRAC channel gating. Proc Natl Acad Sci U S A 115, E5193-E5202.

Zweifach, A., and Lewis, R.S. (1995). Rapid inactivation of depletion-activated calcium current (ICRAC) due to local calcium feedback. J Gen Physiol 105, 209-226.